# Combined Use of a Bacterial Consortium and Early-Colonizing Plants as a Treatment for Soil Recovery after Fire: A Model Based on Los Guájares (Granada, Spain) Wildfire

**DOI:** 10.3390/biology12081093

**Published:** 2023-08-05

**Authors:** Marla Niza Costa, Tatiana Gil, Raquel Teixeira, Ana Sofía Rodrígues dos Santos, Inês Rebelo Romão, Cristina Sequero López, Juan Ignacio Vílchez

**Affiliations:** 1Instituto de Tecnologia Química e Biológica (ITQB)-NOVA, iPlantMicro Lab, 2780-157 Oeiras, Portugal; mniza@itqb.unl.pt (M.N.C.); tgarridogil@itqb.unl.pt (T.G.); ral.teixeira@campus.fct.unl.pt (R.T.); anasofiasantos@itqb.unl.pt (A.S.R.d.S.); ines.rromao@itqb.unl.pt (I.R.R.); 2GeoBioTec, Department of Earth Sciences, NOVA School of Sciences and Technology, Universidade NOVA de Lisboa, Campus de Caparica, 2829-516 Caparica, Portugal; cs.lopez@fct.unl.pt

**Keywords:** fired soil, microbial communities, emerging colonizers, vegetable coverture symbiosis, microbial nutrient cycling

## Abstract

**Simple Summary:**

In recent years, seasonal wildfires have become more frequent and intense. The impact on soils can cause a critical loss for the ecosystems, and the early stages after the events are especially relevant to their recovery. Emerging plants enhance soil stabilization, and the presence of a healthy microbial population is important for aggregating soil particles. In this present study, we analyzed the changes in the microbial population in the early stages, and the relevance of the disappearance of these species due to wildfires. Here, we defined how a treatment combining early colonizing plants and a consortium of bacteria can enhance soil structure and accelerate the soil recovery process.

**Abstract:**

During 2022, intense heat waves, together with particularly extreme dry conditions, created a propitious scenario for wildfires, resulting in the area of vegetation consumed in Europe doubling. Mediterranean countries have been particularly affected, reaching 293,155 hectares in Spain, the worst data in the last 15 years. The effects on the vegetation and the soil are devastating, so knowing the recovery factors is essential for after-fire management. Resilient microorganisms play a fundamental role in rapid nutrient recycling, soil structure, and plant colonization in fire-affected soils. In this present work, we have studied emergent microbial communities in the case of the Los Guájares (Granada, Spain) fire, one of the most extensive of the year, to evaluate their role in the recovery of soil and vegetation cover. We aim to discern which are the main actors in order to formulate a new treatment that helps in the ecosystem recovery. Thus, we have found the relevant loss in phosphorous and potassium solubilizers, as well as siderophores or biofilm producers. Here, we decided to use the strains *Pseudomonas koreensis* AC, *Peribacillus frigoritolerans* CB, *Pseudomonas fluorescens* DC, *Paenibacillus lautus* C, *Bacillus toyonensis* CD, and *Paenarthrobacter nitroguajacolicus* AI as a consortium, as they showed most of the capacities required in a regenerative treatment. On the other hand, the microcosm test showed an enhanced pattern of germination of the emerging model plant, *Bituminaria bituminosa*, as well as a more aggregated structure for soil. This new approach can create a relevant approach in order to recover fire-affected soils in the future.

## 1. Introduction

Climatic conditions in Europe, especially in the southern and Mediterranean regions, are becoming increasingly extreme. This situation is especially notable due to the increase in temperature, as well as in the intensity and duration of heat waves, or the consequent hardening of seasonally dry periods. On average, the 10 hottest years recorded have occurred since 2010, with 2022 being the sixth warmest year on average in the historical series [1]. Moreover, the trend in summer rainfall in Southern Europe since the end of the 20th century has been decreasing significantly by approximately 20 mm per decade, reaching 90 mm in the case of the Iberian Peninsula [2]. This situation has generated a very dangerous scenario. Although most are still classified as intentional or accidental, natural origin fires are also becoming more intense in terms of burned area. The year 2022 has been especially difficult in Europe, recording 785,000 hectares (ha), representing more than double the number of hectares in the 2006–2021 period in total [3]. Countries such as France, Croatia, or Portugal have registered increasingly intense fires and a larger affected area. However, in Spain, these conditions have worsened the most: in 2022 alone, up to 60 sources of large fires were registered (those larger than 500 ha), the highest number in the last decade, for a total of 309 thousand hectares burned, registering the worst data in the last 15 years [3,4].

Among the most harmful fires of 2022 in Spain, both by number of hectares and by the natural value of the affected area, is the fire of “Los Guájares” (Granada). Between September and October, 5200 hectares of scrubland and forest were burned, reaching a total perimeter of 150 km, the 12th by area of 2022 in Spain [3]. This fire also threatened agricultural land and villages in the municipalities of Los Guájares, El Valle, Albuñuelas, El Pinar, and Vélez de Benaudalla (Granada). According to the Spanish Forest law (21/2015, July 20th), the change in use of fired places is banned for at least 30 years, and we must contemplate the necessity to prepare a specific recovery and reforestation plan for the affected land. However, administrative processes can delay such actions for several years, and no specific recovery techniques are included, so most of the time there is not a clear process to follow. Therefore, it is very important to assess the impacts in terms of soil quality, burned vegetation, and processes that may aggravate the situation, such as erosion or seasonal drought, which are more and more frequent phenomena in the region.

Apart from the obvious effects on fauna and flora, the degradation and destruction of soils is one of the most serious issues for the Mediterranean ecosystem due to the extremely slow soil regeneration ratio [5,6,7,8,9,10]. This component is especially critical because it should serve as support to the recovery and re-colonization of the affected area. To briefly contextualize, low-intensity fires used to increase nutrients in the soil [5]. However, in both low- and high-intensity fires, the loss of essential nutrients is the most accentuated trend in the medium term, especially in autumn–winter typical in the Iberian Peninsula, where the wind and rain erode and easily wash away the bad aggregated remaining soil, leading to even more considerable nutrient losses [11,12,13,14,15,16,17,18]. One of the most obvious changes is the carbon content. Thus, owing to the burning of plant matter and soil organic matter, the loss of carbon can contravene the soil structure by causing compaction [12,13,19]. Although each case is different and burning intensity plays an important role, in general, soils are enriched in nitrogen released by vegetation cover [6,20,21]. Regarding sulfur and phosphorus presence after fire events, they are used to increase in their mineral forms [6,14,20,21,22]. Finally, the initial increase in soil pH may have a positive impact on soil accumulation of Fe, Mg, and Ca complexes and some other micronutrients; however, not in all fires increase the soil pH but decrease it, showing this factor is highly dependent on the basal conditions in each wildfire event [12,13,23]. In this sense, the most intense fire events in the mountain Mediterranean ecosystems, due to the type of vegetation and soil characteristics (low depth, poor basal soil structure, small aggregate size, and low stability), favor nutrient washing [6,20,21,24,25].

Moreover, this soil structure may face a remarkable deterioration. Intense wildfires that a region is more continuously suffering have a detrimental effect on the physical properties of soil, especially by the consumption of soil organic matter. Since soil organic matter contains sand, silt, and clay particles, a loss of soil organic matter results in a loss of soil structure [12,19,23,26]. This leads to a reduction in the porosity, decreasing water infiltration rates, and water storage capacity. In addition, it must be considered that the depth of the ground is a key factor in cushioning the effects of fires, making the degree of impact very diverse. In soils with lower depths, as in our case study, it is very important to evaluate the degree of impact to determine its recovery potential.

Furthermore, it is important to point out that fires are going to dramatically change the microbiota associated with the soil [27,28,29]. The loss of these microorganisms not only affects biodiversity and the networks of interactions with plants or animals, but also their structures, such as filaments or biofilms, that are very relevant as adhesive agents aggregating particles and improving the soil structure, as some authors reported for this region [9,29,30]. Hence, among the most common bacterial genera and species referenced in soils subject to different fire intensity, we can find *Pseudomonas* spp. (as *Pseudomonas fluorescens*), *Arthrobacter* spp., *Burkholderia* spp., *Cohnella* spp., *Massilia* spp., *Blastococcus* spp., *Microvirga* spp., *Stenotrophomonas* spp., *Bacillus* spp., and *Clostridium* spp. All these species are present in Mediterranean soils, and may serve in soil aggregation, as most of them are able to produce biofilms [27,28,29,30,31,32]. These microbial structures and activities can help in moisture retention [28,31,33], as well as help in nutrient cycling, critically influencing the availability of the main nutrients (fixing N, solubilizing P, K, S oxidizing) and micronutrients (siderophores and chelators for Mg, Ca, or Fe) for soil organisms and plants [34]. However, it is relevant to highlight how previous studies have reported that changes in the microbial communities may take years or even decades to recover, conditioning their role in soil [33,35]. Thus, they are highly relevant in the recovery processes and re-colonization of burned ecosystems, being useful as a fire impact assessment measure. Surviving microbes are consequently going to be critically important in how soils and ecosystems are going to recover, as well as in the time required for these processes.

The recovery of ecosystems affected by fire can take more than 10 years, reaching even more than 50 in Mediterranean regions [7,9,23,36,37,38]. Hence, these ecosystems are especially delicate, as they are totally exposed, and erosion phenomena can aggravate the situation, slowing down recovery even more. For this reason, it is very important that rapidly colonizing species (ecologically known as ‘type r’) establish themselves and begin to fix the remaining soil, at the same time that they begin to contribute organic matter [37,38,39]. In this sense, different communities of lichens, mosses, as well as fleeting plants (usually herbaceous), are the first to colonize and begin the process [35,39,40]. However, as most seeds will have been disabled by fire, only those with a fire-adapted strategy will be able to emerge quickly, along with those that can be transported by animals [40,41]. In Mediterranean ecosystems, some plant species, such as *Coris monspelliensis*, *Bromus* sp., or *Lathyrus* sp., have been reported to rapidly emerge after fire events [42,43]. For them to be successful, their interaction with beneficial bacteria is very relevant, since bacteria can expand root systems and improve growth in soil that may still be hostile to plants [44]. Among emerging plants, legumes, such as in the *Lathyrus* sp. case, could be the most interesting because the nitrogen fixation in their roots can accelerate the re-colonization of more demanding plant species, thereby reducing recovery times [45,46]. This strategy may make a lot of sense in the Mediterranean regions, since many of the legumes in these regions are also resistant to intense drought and high-temperature stress conditions [46,47]. In this present study, we analyzed the early response patterns in the “Los Guájares” wildfire as a good model for Mediterranean ecosystems. In the Mediterranean region, the common mismanagement of many forest masses, as well as the gradual abandonment of rural areas, whose populations have traditionally played an important role in the use and care of forests and scrub areas, aggravate the situation of these ecosystems [48,49]. Under these conditions, wildfires are expected to grow in intensity and number, which would worsen the conditions of ecosystems and their resilience. Therefore, it is necessary to improve management systems and anticipate palliative and recovery measures for affected soils and ecosystems. Traditionally, hillsides are stabilized through the use of dikes and terraces, but without cohesion and aggregation, it is only a matter of time before the material is lost [50,51]. Microorganisms and plants are considered the best performers in the fixation of nutrients and soil materials. In this sense, plants are able to fix and aggregate soil due to the expansion of their root system, especially through radicles and hairy roots, but also because of the compounds they exude, which can help to stick to nearby particles [52,53,54,55]. It has been described how both fungi and bacteria participate in soil aggregation, improving its structure, excreting polysaccharides, forming biofilms, or developing matrices by the segregation of fibrous compounds or the expansion of their structures [56,57,58,59]. This perspective made us focus on the knowledge and use of emerging plants and microbial population changes as main factors in order to speed up the soil recovery process.

This was performed in terms of emerging plants and microbial dynamics. Here, we worked with the hypothesis of using local microbiota that were lost because of the fire event, together with early-colonizing plants appearing in the area of the case study, as a combined treatment may help to accelerate soil recovery after fire events. Thus, we identified that about 70% of the emerging plants in all locations correspond to *Bituminaria bituminosa* (C.H.Stirt.), a very resistant legume that also fixes nitrogen in the soil, so we decided to use it as a model plant to predict the relevance of the early-colonizers after fire events in this case study. Furthermore, the stabilization of soil and nutrients are a priority to avoid even more degradation in the affected ecosystems. With them, the establishment and development of emerging plants should be enhanced, also improving soil retention and ecosystem recovery. With all this, we decided to evaluate the bacterial communities in areas affected and not affected by the fire, both in soil and associated with our model plant, in order to prepare a consortium of the most relevant ones. Here, we prepared a microcosm approach with different treatments (mock, consortium, plant, and plant-consortium) and conditions (mimicking burnt soils and unburnt soils). This design allowed us to identify evolution and soil recovery patterns in each case, indicating to us which one may have more potential to serve as an initial step towards new treatments to deal with fire effects in Mediterranean ecosystems in the future.

## 2. Materials and Methods

### 2.1. Sampling

The sampling process was prepared in the region of ‘Los Guájares’ (Granada, Spain), which is located in the climatic area of Sierras de Tejeda y Almijara (mid-mountain Mediterranean). The locations are considered part of the same type of vegetation unit, repopulated pine forest (*Pinus halepensis*), with patches of sclerophyll vegetation and Mediterranean grassland-scrub. All soils sampled were identified as cambic calcisol. The burnt soils were selected in locations clearly affected by the fire (non-border or transition areas), and the unburnt in the closest clearer non-affected locations (about 200–600 m in distance). Due to the low average depth of the soils in the area, we decided to take samples only at two different depths, 0–5 cm (more affected and richer in ashes) and 5–10 cm (less affected section). Each sample consisted of 60–70 g of soil collected from three locations (Table 1 and Figure 1). Moreover, as the most common emerging plant for all locations, pitch clover (*Bituminaria bituminosa* (L.) C.H.Stirt.; about 70% of all emerging plants species at sampling time) was sampled as a model plant in fire-affected soils. For comparison, seedlings of this species at similar developmental stages were also collected from unburnt soils. All samples were collected during the first week of January 2023, approximately four months after the fire event, and preserved at 4 °C until processing.

### 2.2. Soil Characterization

The characterization of the soils (burned and unburned) at different depths (0–5 cm and 5–10 cm) was performed in order to evaluate the characteristic that may have an influence over the availability of nutrients, as well as the different statuses and aggregations of the sampled soils. This study was carried out by the Soil Analysis Laboratory (Laboratório de Análise de Solos) of the National Institute of Agricultural and Veterinary Research (Instituto Nacional de Investigação Agrária e Veterinária, INIAV) in Lisbon (Portugal). This characterization included textural evaluation (PE-005-LQARS/LAS), pH in water (ISO Standard 10390, organic matter by wet way (by sodium dichromate method/EAM UV/Vis (PE-017-LQARS/LAS); using the correction 1.29 for incomplete oxidation of dichromate organic matter), extractable phosphorus (by P-AL Ammonium lactate method (Egner-Riehm)/ICP-OES (PE-021-LQARS/LAS), extractable potassium (Egner-Riehm/ICP-OES (PE-021-LQARS/LAS), extractable magnesium (by 1 M ammonium acetate (pH = 7) method/EAA with flame (PE-008-LQARS/LAS), limes (According to the Table of Quelhas dos Santos), extractable calcium (by 1 M ammonium acetate (pH = 7) method/EAA with flame (PE-008-LQARS/LAS), extractable iron (by Acidic Ammonium Acetate-Ethylene Diamine tetra Acetic Acid (AAAc-EDTA) (Lakanen) method/EAA with flame (PE-016-LQARS/LAS)), and total nitrogen (by dry Combustion—Elemental Analysis (ISO 13878)). Moreover, we analyzed the electroconductivity of the soil samples by using an EC-meter YIERYI EC-8801 (Shenzhen, China), and their percentage of water content (as in Li and Wang, 2014 [58]; by weight difference after 7 days incubation at 60 °C).

### 2.3. Isolation, Identification and Analysis of the Culturable Populations in Ashes, Soil and Roots

The characterization of the culturable populations was carried out in order to better understand the effects of fire on soil, and how early stages after the event evolve. These stages are critical in nutrient content in soil, aggregation ratio of the soil particles, or emerging plant spreading. Therefore, we decided to study how the microbial population changed in order to identify patterns we can use later on to design our proposed treatment, based on presence/absence and representativity of key role strains. Later on, bacterial traits over nutrients, soil aggregation, or beneficial effects on plants will complete this evaluation.

Thus, soil samples, a mix of ashes, and roots (0.5 g) were grinded and processed to isolate culturable bacterial populations. The roots of pitch clover seedlings were surface-sterilized to isolate colonizing strains (endophytic or epiphytic resistant by biofilm structures), avoiding any other external contamination [60]. Roots were first surface-sterilized by immersion in a 20% bleach solution for 3 min, followed by immersion in 70% ethanol for 5 min, and finally washed three times with sterile double-distilled water. For soil and ashes, the initial amount was sieved to avoid rocks, and then diluted in 0.45% NaCl solution. Here, the samples were then serially diluted in 0.45% NaCl sterile solution before drop-by-drop plating on LB agar medium (per liter: NaCl, 10 g; yeast extract, 5 g; tryptone, 10 g; agar, 15 g). The population for each location was counted (colony-forming units [CFUs]) and normalized by dry weight (in mg) of the original sample. Thereafter, morphologically different colonies (as described by the American Society for Microbiology; [61]) were selected, isolated, and purified. Pure cultures were preserved in 40% glycerol at −80 °C.

For strain identification, genomic DNA was extracted from pure-cultured colonies using the heat shock method [62]. After assessing the quality using Nanodrop spectroscopy (Thermo Scientific™ NanoDrop™ One Microvolume UV-Vis Spectrophotometer, Waltham, MA, USA), the hypervariable V5–V8 region of the 16S rRNA gene (approximately 700 bp) was amplified with the universal primers 779F (5′-AACMGGATTAGATACCCKG-3′) and 1392R (5′-GGTTACCTTGTTACGACTT-3. PCR was carried out with NZYTaq II 2x Master Mix by Nzytech, with the following running conditions: 2 min initial denaturation phase (95 °C); then 40 cycles of 3 s denaturation (95 °C), 30 s of annealing (52 °C), and 2 min elongation (72 °C); and a final 7 min extension phase (72 °C). The amplicon was assessed by electrophoresis (1% agarose) using a Mupid^®^-exU System (Takara Bio, Kusatsu, Japan) at 135 V, followed by a transilluminator (UVIvue™; Dutscher, Issy-les-Moulineaux, France).

Samples were sequenced using GENEWIZ (Leipzig, Germany), and then, the sequences were submitted to the National Library of Medicine BLAST database to identify the strains (>98% similarity). Strains were considered as ‘Unidentified’ when we were not able to amplify the 16S rRNA or didn´t have quality enough for sequencing after several attempts. Finally, a phylogenetic tree was prepared with the sequences of the identified strains by aligning in ClustalX2 (v2.0; University College Dublin, Ireland) and visualizing in the iTol drawing tool (https://itol.embl.de/itol.cgi; accessed on 10 May 2023) [63].

### 2.4. Nutrient Cycling Skills

The study of the nutrient-cycling traits in the microbial communities was very relevant to us in order to avoid the usual washing of nutrients occurring in Mediterranean ecosystems after a fire (autumn rains), due to the loss of soil structure. Normally, these nutrients are in a mineral form after a fire, which makes them inaccessible to plants, hindering their emergence and spreading. Microbes can play a role in changing the form of certain nutrients (such as P and K solubilization or sulfur oxidation) to make them more accessible, as well as increase their availability in soil, facilitating plant development. With the study of populations in burnt vs. unburnt soils, and the screening for microbe skills, we can also elucidate how the remaining microbes perform in affected soils and compare if they are enough to carry on these processes, or conversely, if the part of the population that was lost shows better performance. Therefore, this helped us to decide which strains to incorporate in the proposed consortium treatment.

#### 2.4.1. Screening of Nitrogen Fixing Activity

The nitrogen fixation was screened in our strains by following indications of Sulistiyani and Meliah, using Bromothymol Blue (BTB) (100 mg/L) as a color indicator method [64]. Strains were grown in solid Jensen’s medium (per liter: 20 g of sucrose, 1 g of K_2_HPO_4_, 0.5 g of MgSO_4_, 0.5 g of NaCl, 0.1 g of Fe_2_SO_4_, 0.005 g of Na_2_MoO_4_·2H_2_O, 2 g of CaCO_3_, and 15 g of agar) for 7 days, when the color of the medium changed to dark blue. The colored halos were measured, and the relative nitrogen fixation ability (RnfA) was calculated with the following formula: RnfA = (total zone size − colony size)/colony size. This same formula was also applied for the rest of the halo-related tests. Strains with positive results in this test were grown again in liquid Jensen’s medium with BTB (30 °C, 150 rpm) for 3 days to measure the absorbance produced at 640 nm in a microplate reader plate system [65,66]. A nitrogen-equivalent standard curve was prepared by increasing concentrations of NH_3_.

#### 2.4.2. Phosphate Solubilization

For the screening of phosphate-solubilizing bacteria (PSB), we followed Zheng and collaborators, as well as Reyes, Valery, and Valduzin’s methodologies, with slight modifications [67,68]. Here, we prepared NBRIP agar medium (per liter: 10 g of glucose, 5 g of Ca_3_(PO_4_)_2_, 5 g of MgCl_2_·6H_2_O, 0.25 g of MgSO_4_·7H_2_O, 0.2 g of KCl, 0.1 g of (NH_4_)2SO_4_ and 15 g of agar), and placed a drop of 10 µL of an overnight growth culture (in LB; 28 °C, 150 rpm) of each strain that was previously centrifuged (5 min; 14,000 rpm) and resuspended in sterile solution of 0.45% NaCl. After incubating for 5–7 days at 28 °C, strains that developed discoloration halos were considered PSB. The diameter of the halos was also measured. For quantification, using 96-well plates, strains were grown in NBRIP liquid medium as indicated before, and after three days, the medium turbidity was measured at 600 nm, where a lower value indicated the solubilization of phosphate in the medium. The plates were centrifuged (10 min, 4000 rpm) and the supernatant was transferred to a fresh plate and mixed with 3, 5:1 (*v*:*v*) vanadate-molybdate reagent. The plate was then incubated in the dark for 10 min and the absorbance was measured at 420 nm. A phosphate standard curve was constructed using anhydrous KH_2_PO_4_.

#### 2.4.3. Potassium Solubilization

The screening for potassium-solubilizing bacteria (KSB) was prepared according to Etesami, Emami and Alikhani, and Rajawat and collaborators, with minor modifications [69]. Thus, similar drop system to PSB screening was prepared but using Aleksandrov medium (per liter: 5.0 g of glucose, 0.5 g of MgSO_4_·7H_2_O, 0.1 g of CaCO_3_, 0.006 g of FeCl_3_, 1.5 g of K_2_HPO_4_, 1.5 g of KH_2_PO_4_, 3.0 g of potassium aluminum silicate (mica) and 15 g agar; pH 7.2), amended with BTB (100 mg/L). The plates were incubated for 3–5 days at 28 °C. For positive results, yellowish halos around strains deposited in the medium were detected, and their diameters were recorded. Then, for quantification of the available solubilized, we used Rajawat formula for the standard curve. Additionally, we cultured the strains in Aleksandrov liquid medium in a 96-well plate (150 rpm for 3 days), and we measured the absorbance at 430 nm to compare and adjust the results.

#### 2.4.4. Sulfur-Oxidation

The screening of the sulfur-oxidizing bacteria (SOB) were screened based on the methodology of Hidayat, Saud, and Samsudin, with some modifications for our type of samples [70]. Briefly, the strains were grown in thiosulfate mineral medium (TSM) (per liter: 1.5 g of K_2_HPO_4_, 1.5 g of KH_2_PO_4_, 0.4 g of NH_4_Cl, 0.8 g of MgCl_2_·6H_2_O, 0.1 g of CaCl_2_·2H_2_O, 10 g of Na_2_S_2_O_3_·5H_2_O and 15 g of agar; pH 7.5), amended with 0.01 g of bromocresol purple. A drop of each culture strain was deposited in the plates and incubated for 14 days at 28 °C. The discoloration halo around each strain was considered as positive. For quantification, cultures were prepared in TSM liquid medium and grown for 14 days at 28 °C and 160 rpm. The supernatant was separated from the cells by centrifugation (10 min, 4000 rpm) and vigorously mixed 1:1 (*v*:*v*) with barium chloride (BaCl_2_) solution (10% *w*/*v*) until a turbid solution of barium sulfate (BaSO_4_) was produced. The absorbance was measured at 480 nm. A standard sulfate solution was prepared by dissolving potassium sulfate (K_2_SO_4_) in a BaCl_2_ solution.

#### 2.4.5. Siderophores Production

Screening for siderophore producers was performed according to the method described by Arora and Verma, with minor modifications. Here, drops of each strain were deposited in Blue Agar Chrome Azurol Sulfonate (CAS) medium plates [71,72]. A colored halo around the strain was considered as positive. For quantification, a regular LB liquid culture of each strain was prepared, and after centrifugation (10 min, 4000 rpm), 100 µL of the supernatant was mixed with 100 µL of CAS reagent in a 96-well plate. A mixture of LB broth and CAS reagent was used as the control. After 20 min of incubation, absorbance was recorded at 630 nm. Siderophore production was calculated as the percent siderophore unit (psu) according to the following formula: psu = (absorbance control – absorbance sample) × (100 × absorbance control).

### 2.5. Biofilm Production

Biofilm production of all the strains can help in peds aggregation and soil stabilization, so it was evaluated in our collection. For this, we followed the method described by Coffey and Anderson [73], with slight modifications. In brief, 200 µL of a culture in LB medium (starting with a 0.05 OD_550nm_ culture of each strain) was incubated for 24 h in a 96-well plate at 28 °C and 150 rpm. After washing the planktonic bacteria from the wells with tap water, 200 µL of 0.5% crystal violet solution was added to the wells to stain the adhered structures, and the plates were incubated for 20 min at room temperature. After washing the excess crystal violet, the biofilm structures were solubilized with 30% glacial acetic acid solution for 20 min, and the solution was measured at 550 nm.

### 2.6. Plant Growth Promoting (PGP) and Stress-Tolerance Enhancing Traits

The production of several compounds by bacteria, such as phytohormones and antioxidants, is relevant for plant growth as well as for plant responses to stressful conditions. The establishment of emerging plants in burnt soils has to face a hostile environment, so beneficial associations with microbes producing these compounds improve their adaptation. In this study, we tested the production of auxins (secondary root production), 1-aminocyclopropane-1-carboxylate (ACC) deaminase (control of harmful effects of ethylene), and antioxidants (cleavage of reactive oxygen species) in the isolated strains.

Thus, beginning with auxin production, we followed the method described by Ambrosini and Passaglia [74]. Thus, using a 96-well plate, 200 µL of LB broth supplemented with tryptophan (0.5 g/L) was inoculated with a 0.05 OD_600nm_ culture of each strain. Thereafter, the plate was incubated for 48 h at 28 °C and 150 rpm, and then centrifuged (30 min, 4000 rpm). The supernatant (100 µL) was transferred to a fresh plate and mixed with 100 µL of Salkowski reagent (0.5 M of FeCl_3_ and 35% HClO_4_). Finally, after a 30 min incubation at room temperature in the dark, the absorbance was measured at 530 nm. The values were determined as indole-3-acetic acid (IAA) equivalents with respect to a calibration curve.

The production of 1-aminocyclopropane-1-carboxylate deaminase (ACCd) is another key PGP activity that we studied. For this, we followed the ninhydrin method, as indicated by Li and collaborators [75]. Here, in a 96-well plate, strains were grown in M9 medium (Merck) supplemented with 3 mM of ACC. After 24 h in incubation (28 °C and 150 rpm), the growth of the strains was measured at 600 nm. On the other hand, one of the PGP skills that is more important in stressful conditions, such as burnt soil, is the antioxidants production. For this determination, we applied the thiocyanate method for antioxidative activity described by Takao and collaborators [76], with minor modifications. Briefly, a sample of the centrifuged overnight cultured strain (200 μL) was mixed with 200 μL linoleic acid solution (25 mg/mL in EtOH), 400 μL of phosphate buffer (per liter: 20.214 g of Na_2_HPO_4_·7H_2_O, 3.394 g of NaH_2_PO_4_·H_2_O; pH 7.0), and 200 μL of distilled water in a 1.5 mL microtube. The tubes were kept at 40 °C in the dark for 10 min. After this incubation, 100 μL was mixed with 3 mL of 75% EtOH, 100 μL of NH_4_SCN solution (0.3 g/mL in distilled water), and 100 μL of ferrous chloride reagent (2.45 mg/mL of FeCl_2_ in 3.5% hydrochloric acid). Finally, after 3 min of incubation at room temperature, absorbance was measured at 500 nm. Trolox (6-hydroxy-2,5,7,8-tetramethylchroman-2-carboxylic acid) was used as standard, according to Shafekh and collaborators [77].

### 2.7. Microcosm Test

The microcosms were designed as a way to interpret the process that happens in natural soil after our treatments out of field conditions. Due to legislation, we could not use the amount of natural soil required here, as well as it was not possible to prepare these tests in the affected soils. Here, we decided to use a poorly aggregated soil (sandy soil) in order to detect the degree of aggregation that our treatments may cause. The comparison is not direct to the natural soil conditions, but any aggregation degree detected here may indicate a good performance in low-aggregated types of soils, as happen in most of the soil after fire. Thus, we prepared 4 L open trays with draining holes, and a depth of 12.5 cm. They were filled 1 cm deep with a drainage material (claystone) and 9 cm of a 16-mesh (1.19 mm Ø) sieved natural sandy soil (AGRO-TECH Campus de Oeiras, Portugal) that was tyndallized to reduce the microbial load, mimicking the fire effect in soil. In parallel, another set of 2:1 (*v*:*v*) mix of the same natural sandy soil and pinewood and ashes from a burnt area in Los Guájares (approximately 0.75 L/microcosm), were prepared in order to simulate the conditions of the sampling points (Figure 2a,b).

Thus, the conditions applied were (i) soil without treatment (control), (ii) soil treated with bacterial consortium (SynCom), (iii) soil with *B. bituminosa* seeds, and (iv) soil treated with bacterial consortium and *B. bituminosa* seeds (Cantueso Natural Seeds, Córdoba, Spain) (Figure 2c). For bacterial inoculation with the selected strains, synthetic community (SynCom) was prepared following the indications of Schmitz and collaborators, with slight modifications [78]. Here, the strains were individually grown overnight in LB, and then centrifuged (10 min, 10,000 rpm) and adjusted to 10^8^ CFUs/mL in a 0.45% NaCl saline solution before preparing a balanced mixture of them. For bacteria-free treatments, the same sterile saline solution was applied. Seeds were scarified and surface-sterilized (1 min in 20% bleach, 3 × ddH_2_O washing) before being buried in the top 4 cm of the microcosm. All treatments were maintained in a relative humidity of 30–40% by regular water spraying to avoid including stressful effects, and to mimic the soil conditions in the model area. In parallel, the same set of treatments was tested without ash in the soil mix as control conditions. The microcosms were kept in a greenhouse for 40 days to avoid environmental effects that could affect the evaluation of the treatments.

To evaluate and compare the aggregation level achieved by the soils under the different treatments, a granulometric characterization and an image-based slake test (Slakes software v.BETA, University of Sydney—Australia) were performed [79,80]. Moreover, a visual evaluation of the soil fragments/peds was carried out at the end of the treatment (see Section 2.8). Every 10 days from the beginning of the test, the change in the electro-conductivity (EC) and pH of the soil were determined by using the probes described previously in this section. The determination of N, P, and K was performed by using a Soil Integrated Sensor 4001-BXSZD (Walfront). Additionally, we evaluated the germination rate (%) for *B. bituminosa* under each condition. As a reference, we evaluated the germination rate in vitro by using Magenta boxes and wet paper [47]. Here we treated the seeds with each candidate strain, as well as with the whole consortium, using a set of seeds treated just with water as control. For this, after surface-sterilizing the seeds (incubation in 70% ethanol for 5 min, followed by three washes with double distilled water), they were incubated for 6 h in a 10^8^ CFUs/mL solution. Finally, for plants germinating in the microcosm, we also evaluated the degree of development (height and biomass in dry weight) and root architecture (length and number of secondary roots), as indicated in Niza and collaborators [47].

### 2.8. Microscopy

To analyze the structure, we used a Bresser Digital USB Dst-1028 microscope (Bresser GmbH, Rhede, Germany) to take images at 10–20× magnification, with which we were able to evaluate the degree of aggregation of the soil peds under the different treatments.

### 2.9. Statistics and Analyses

The statistical analyses were performed in Prism (v9.0.0; GraphPad Software) using Student’s *t*-test or two-way ANOVA (with Tukey’s post-test and Šidák corrections) for pairwise and multi-group comparisons, as required. The minimal significance level was set at *p <* 0.05. The graphs were analyzed using Prism and Excel 2019. As a biodiversity analysis, we used Shannon index, Simpson index, and evenness (Virtue online tools—virtue.gmbl.se/english-content/biodiversity-calculator, accessed on 10 May 2023). Meanwhile, population analysis was performed with the EVenn tools [81]. Finally, the PCA analyses were prepared with the Principal Component Analysis Calculator from Statistics Kingdom (statskingdom.com/pca-calculator.html). In all the cases, the tools were visited in May 2023.

## 3. Results

### 3.1. Soil Characterization

The characterization of the soil samples was carried out both in burned and unburned areas, as well as in superficial (0–5 cm) and deep (5–10 cm) samples (Table 2). The reports and data presented in this study are openly available in FigShare [82]. Thus, the recorded pH values are generally alkaline and quite similar between samples, being higher than 8 in all cases. With some exceptions, unburned soils were slightly less alkaline, as were superficial samples. Conversely, the samples were generally not very electrically conductive (EC), indicating a low salt content. The EC was generally slightly lower in the deep samples. All samples, both from the top part of the soil and from the deep part, showed a higher EC value in the burned soils than in the unburned ones. Regarding the organic matter (OM) content, as expected, this fraction was lower in the deep samples as well as in the unburned samples. In surface samples 1 and 2, more OM was detected in burnt locations, which is not usual (emerging plants and new exudates and organic compounds may be apported). On the other hand, we were able to detect notable differences in the nutrients analyzed. Beginning with total nitrogen (N) detected, the top soils showed in general more percentage of N than in the deep ones, and, except in sample 1, there was not a significant difference between the burned and unburned samples. Considering the deep samples, the burned ones showed a slightly higher concentration of N, except in sample 3, which was more balanced. In the case of phosphorus (P), except in sample 3, both on the top and deeper soil samples, a greater quantity was found in the burned soils. Among these samples, the highest amount of P was detected in superficial samples. For potassium (K), with the exception of the samples from point 3, which were somewhat more balanced, a higher amount was found in the top soil samples than in the deep ones, and it was also higher in burned soil samples than in deep ones.

In the case of magnesium (Mg), the amounts detected were slightly lower in the deep samples, being found significantly less in unburned soils than in burned ones, reaching between two and six times more Mg. The quantification of calcium (Ca) showed that, excepting the samples from point 2, which was slightly more balanced, a greater quantity was found in the top soil samples than in the deep ones, which is also the trend in burned soils with respect to the unburned soils. Finally, the amount of iron (Fe) detected was slightly lower in deep samples, and much higher in burned soils than in unburned ones, where a Fe-enrichment between two and seven times was found.

### 3.2. Population Analysis: Differences Caused by Wildfire Events

A total of 56 strains were isolated from the roots of *B. bituminosa* and two depths in soil (0–5 cm, and 5–10 cm) from places affected by fire and the closest non-affected locations (Table 3, Figure 3). Moreover, a mix of ashes was also analyzed. All sequences were submitted and are accessible in GenBank through the accession number OQ971766-OQ971803. However, not all strains were able to grow once we tried to prepare pure cultures, and it was not possible to correctly identify one other group of strains (about 30%, where 40% were visually identified as yeasts). The isolates were similarly distributed along the three locations (18, 16, and 14 strains, respectively) with certain internal coherence when attending to the species founded by the isolation condition. Moreover eight strains were isolated from the mix ashes. About 30% of the strains belonged to the family Bacillaceae, 20% to Pseudomonadaceae, and 9% to Micrococcaceae. The latter was particularly different to the rest of the isolates, providing unique strains for the collection (Table 4). Considering the population distribution, on average, the amount of CFUs/mg of ashes was about 7.15 × 10^4^, with *Sphingobacterium* sp. strains Ashes5 and Ashes6 (72%) and Acinetobacter calcoaceticus (12%) as the more prevalent strains (Appendix A).

Attending to the main conditioners analyzed, 22 strains were isolated from burnt places and 26 from unburnt ones. In a general way, 32 strains in the collection were found in roots emerging in these regions, meanwhile, regarding the soil samples, 32 were found in the top part of the soil, and 27 in the deeper soils. Thus, 15 strains were present in roots and in soil, independent of the depth (27%), and 12 were only isolated from soil (21%). Among this last group, only six strains were represented in both depths. Considering the fire effects on the culturable population (Appendix A), samples from burnt locations, in general, they contained about 3.5 times more colony-forming units (CFUs) per milligram of soil dry weight (DW). In both types of locations, roots showed 10 times more population than the rest of the soil (about 20 times more when compared to the deepest fraction of soil). However, samples from burnt locations showed significantly more population than the unburnt in roots, top, and deep samples (about 2.7, 8, and 3 times more, respectively). Considering the strains identified in burnt and unburnt locations, in roots we found three common species (*Pseudomonas fluorescens*, *Peribacillus frigoritolerans* and *Pseudomonas koreensis*), but in soils, only one species (*P. fluorescens*) was found in both kind of locations (Appendix A).

As shown in Figure 4, the most prevalent strains in the roots collected from burnt places were found to be *Bacillus mycoides* and *P. fluorescens* (about 45%), meanwhile in the unburnt locations, the main strains detected were *Paenarthrobacter nitroguajacolicus*, *Bacillus toyonensis*, *P. frigoritolerans*, and *Priestia megaterium* (in more than 50% of the samples). In the case of top soil samples, in the burnt locations the strains *Bacillus cereus*, *Bacillus mycoides*, *Exiguobacterium* sp., and *P. fluorescens* (more than 20%) were the most relevant, while in unburnt places, these were *P. koreensis*, *P. nitroguajacolicus*, and *P. fluorescens* (more than 20%). Finally, *B. mycoides*, *Achromobacter spanius*, and *P. fluorescens* (more than 20%) were the most prevalent in the deepest section of the burnt soils, and *P. megaterium*, *Flavobacterium* sp., and *Pseudomonas lini* (more than 30%) in the case of unburnt soils. Here, *B. cereus*, *Achromobacter spanius*, and *P. fluorescens* were present in all types of samples collected from burnt locations. The strain *Paenibacillus lautus* and *B. mycoides* were only present in root and top soil samples; meanwhile *B. cereus*, *P. fluorescens*, *Exiguobacterium* sp., and *P. koreensis* were present in all kinds of sources. Furthermore, in unburnt samples, *P. koreensis*, *P. frigoritolerans*, *Priestia megaterium*, and *P. nitroguajacolicus* were found in all kinds of samples. Moreover, the strain *B. toyonensis* occurred in root and deep soil samples, and *P. fluorescens* in root and top soil samples. Finally, the strain *Pseudomonas brassicacearum* was only present in deep soil samples. Thus, burnt and unburnt locations shared the presence of the strains *P. frigoritolerans* and *P. fluorescens*. However, the strains *B. cereus*, *P. lautus*, *A. spanius*, *Exiguobacterium* sp., *Exiguobacterium mexicanum*, and *Bacillus mycoides* were exclusively isolated from burnt locations and the strains *P. nitroguajacolicus*, *Pseudomonas lini*, *P. nitroguajacolicus*, *B. toyonensis*, *Flavobacterium* sp., *Priestia megaterium*, and *P. brassicacearum*, from the unburnt ones. The strain *P. koreensis* was minimally present in the samples from burnt locations.

In addition, we also prepared an evaluation of biodiversity indexes. Here, we considered the Evenness (E), the Shannon–Weiner index (H) and the Simpson’s index (D), including the diversity (1 − D) and the reciprocal (1/D) modifications. Hence, beginning with the E index, a measure of how similar the abundances of different species are in the community, we found significant differences in all the profiles of the samples, being less abundant in the burnt samples for root and deep soil; meanwhile, they increased for top soil populations. For the H biodiversity index, which measures the order observed within a particular system, populations recorded in samples of burnt soil (top and deep) were shown to be more disordered (less balanced) with respect to those in unburnt soils. Notwithstanding, the populations in roots were shown to be slightly more ordered in the burnt locations. Attending to the D biodiversity index, which evaluates the probability that two randomly selected individuals in the community belong to the same category, we only observed significant differences for deep soil in the burnt locations. In addition, we also evaluated the D modifications, 1 − D and 1/D, adding the consideration of the number of different categories (e.g., species) and the number of equally common categories, respectively. Contrary to the D index, the only significant difference was detected between populations in deep soil samples from burnt locations, being this time lower compared to the values obtained for the population in unburnt locations (Appendix A).

### 3.3. Screening of Nutrient-Related, Structuring and Plant Growth-Promoting Skills

The full collection of strains was initially tested under plate, semi-quantitative methods to discriminate the best candidates (Appendix A), whose skills were properly quantified using 96-well plate, spectrophotometry-based methodologies. Raw data and analysis of all these initial tests presented in this section are openly available in FigShare [83]. Hence, beginning with nitrogen fixation abilities, we found that about 21% of the strains were able to fix nitrogen in plate (a total of 12 strains), with five of them showing a relative activity value above 10 (as *Pantoea agglomerans* Ashes3, *E. mexicanum* BG and *P. koreensis* E-alfa). The amount of nitrogen fixators was similar among samples from burnt and unburnt locations (5 and 6, respectively), with only one strain in ashes. However only one strain of the top producers was coming from unburnt locations. In the profile of the samples, most of them were found in roots (9) and in the top soil samples (8), reducing their presence in the deep soil samples (6).

Focusing on phosphate solubilizing strains, we detected a total of 46 strains (82%) in our samples. A total of 23 strains were isolated from unburnt locations, and 19 from the burnt ones. Half of the strains isolated in ashes (4) were able to solubilize phosphate. Considering the profile of the samples, all of them registered similar numbers (31 in roots, 27 in top soil, and 26 in deep soil samples). Among them, 22 showed relative activity values above 10, and 4 of them even above 20 (as *Acidovorax* sp. Ashes7, *P. nitroguajacolicus* AI, *P. frigoritolerans* EB, *P. nitroguajacolicus* AE), all of them in unburnt locations and ashes. The population of potassium solubilizers among our isolates was about 77% (43 strains), with 35 strains above 10 in relative activity, and 9 above 40 (as *Pseudomonas* sp. Ashes4, *P. nitroguajacolicus* AI, *P. fluorescens* BE, *P. frigoritolerans* AJ, *P. fluorescens* DE). Here, the profile of the samples is more balanced, with 28 in root samples, 24 in top soil, and 23 in deep soil. In terms of distribution in areas affected by fire, we found 24% less potassium solubilizers (16) than in unburnt locations (21). For ashes, 75% of the strains were able to solubilize potassium.

On the other hand, the oxidation of sulfur to accessible forms for plants was addressed in more than 70% of the strains isolated (41 in total). From these, 22 strains were found in unburnt locations, 40% more than in burnt locations (13). In ashes, 75% of the strains were positive for sulfur oxidation (6 strains). Among them, four strains obtained a relative activity above 10 (as *P. nitroguajacolicus* AE, *P. frigoritolerans*/simplex CA, *P. frigoritolerans* AJ, *Flavobacterium* sp. CH), all of them isolated from unburnt locations. The number of strains with this skill was similar in root (26), top soil (22), and deep soil samples (21), showing a slight decrease in prevalence at increasing depths. In the case of siderophore producers, a total of 17 strains in our collection were positive (30%). We recorded more than double of them in unburnt locations (11) than in the burnt ones (5), with only one among the ash samples (12.5%). Among the siderophore producers, 12 strains showed a 1.1 value of relative activity (as *P. koreensis* AC, *P. koreensis* E-alfa, *Exiguobacterium* sp. BD, *P. frigoritolerans*/simplex CA, *P. frigoritolerans* CB, *Flavobacterium* sp. CH, *P. nitroguajacolicus* EA, *P. granadensis* DE, *P. nitroguajacolicus* CI, *B. toyonensis* CD, *P. frigoritolerans* BB, *P. frigoritolerans* EB), most of them isolated from unburnt areas, and 1 strain with a relative value of 7 (*P. fluorescens* CF). Considering the profile, 12 strains were isolated from root samples, meanwhile 10 were isolated from top soil, and 8 from deep soil samples.

Interestingly, we were able to detect a decent amount of biofilm in most of the strains, however, only 24 (43%) were able to produce above a 0.55 (OD_550nm_), referential level of the high-producer strain *Pseudomonas putida* KT2440, which we used as a control for this test. Of these, 18 were isolated in unburnt locations, almost double compared to burnt locations (10). More than half of the strains isolated (5) from ashes were able to produce this level of biofilm. Top producers belonged to *Peribacillus genus* (*P. frigoritolerans* AK (3,60) and *Bacillus toyonensis* CD (2,53)), followed by *Sphingobacterium* sp. Ashes6 (1,78). Attending to the profile of the samples, they were very balanced (18 strains were found in roots, 17 in top soil, and 18 in deep soil).

The auxin production was prevalent among most of the strains in the collection. Considering again *P. putida* KT2440 as the control strain (about 25 µg/mL), 14 strains showed more production of indoleacetic acid equivalents (25%). In ashes, almost half of the strains were high producers (3). Strains producing above this level were more than double in unburnt locations (8) than in burnt ones (3). All the top producers, *P. megaterium* ED (62,09 µg/mL), *Flavobacterium* sp. CH (60,74 µg/mL), and *P. koreensis* AC (50,32 µg/mL), were isolated from unburnt locations. Moreover, eight strains of this group were isolated from unburnt locations and three from burnt ones, the same number as in the ashes. In the sampling profile, the strains happened similarly in roots (10), top soil (10), and deep soil (7) samples. Interestingly, the strain Flavobacterium sp. CH was only present in soil samples. In the case of the ACC deaminase production, in general, all the strains isolated were shown to be able to grow in a media supplied only with ACC as a carbon and nitrogen source. Here, using *P. putida* KT2440 growth (OD_600nm_) as our reference, almost half of the population grew above 0.5 (OD_600nm_), with 13 isolated from unburnt locations, 10 from burnt locations, and 4 from ashes. Considering the profile of the sources, they were shown to be balanced (17 in roots, 13 in top soil, and 14 in deep soil), with five strains only happening in soil samples, and another five only in roots, mainly identified as *P. nitroguajacolicus* and *P. frigoritolerans*, respectively. Although these measurements do not allow us to evaluate the ACCd produced or the efficiency, the strains with more growth (*P. koreensis* AB, *B. mycoides* D, and *P. frigoritolerans* AK, all above 0.76 OD_600nm_) are probably the ones with better performance in the use of this enzyme to control ethylene levels.

Finally, when we analyzed the antioxidant production, we observed that all the strains were in some way able to produce a certain amount. However, only 23 of them (about 36%), were able to produce above 4 mM of antioxidant equivalents. From the strains we were able to identify, only two strains produced above 6 mM (*Exiguobacterium mexicanum* BG and *P. fluorescens* DC), both isolated from burnt samples. In general, the strains isolated from burnt samples and ashes were the ones producing more antioxidants. Considering only the strains producing above 4 mM, 10 strains were isolated from burnt locations, 9 from unburnt locations, and 4 from ashes (50% of the strains from this kind of sample). Moreover, 16 were isolated in root samples, 15 from top soil, and 13 from the deep soil samples. Only two strains were exclusive from soil samples (as *E. mexicanun* BG), and the other two exclusively from roots (as *P. fluorescens* BG and *P. fluorescens* DD), and both only present in fire-affected samples.

In most of the cases, the traits evaluated were similarly prevalent in both kinds of locations; however, K solubilization, sulfur oxidizing, siderophore, biofilm, and auxin production seemed to be more prevalent in strains isolated in unburnt locations. Here, we decided to evaluate the prevalent traits clustering by principal component analysis (PCA). The results (Figure 5) showed a recognizable clustering among the strains by origin, where the cluster containing the strains isolated from burnt places was more closely related to the strains isolated in the mix of ashes. Moreover, the strains isolated from unburnt locations showed a differentiable clustering pattern. Here, we can highlight that the strains isolated in the unburnt locations were more related to the sulfur oxidizing, siderophore, biofilm, and auxin production vectors; meanwhile, the strains isolated from ashes and from burnt locations were more aligned with the vectors of nitrogen fixation or antioxidant production.

### 3.4. Selection of Candidates and Deep Characterization

After the evaluation of the capacities of the isolated strains, we decided to prepare a consortium to evaluate their effect to recover burnt soils. To choose the best candidates, we analysed the best strains in each of the previous assays. Here, we decided to balance the presence of each strain to ensure that each of the abilities was well represented and covered by the consortium strains. Thus, we decided to select six strains: *P. koreensis* AC (antioxidants, auxins, nitrogen fixation [113.2 µg/mL of NH_3_^+^ equivalents], phosphate solubilization [222.7 µg/mL of PO_4_^−^ equivalents], siderophores [49.4 psu]), *P. frigoritolerans*/simplex CB (biofilm [2.53 A_550nm_], P [270.5 µg/mL of PO_4_^−^ equivalents] and K [128 µg/mL of K^+^ equivalents] solubilization, siderophores [56.2 psu], S-oxidizing [9.1 mg/mL of SO_4_^2−^ equivalents]), *P. fluorescens* DC (antioxidants [6 mM], N fixation [137.3 µg/mL of NH_3_^+^ equivalents], *P solubilization* [170.7 µg/mL of PO_4_^−^ equivalents]), *P. lautus* C (P solubilization [188.2 µg/mL of PO_4_^−^ equivalents], ACCd [0.622 A_600nm_]), *B. toyonensis* CD (P solubilization [190.5 µg/mL of PO_4_^−^ equivalents]) and *P. nitroguajacolicus* AI (ACCd [0.650 A_600nm_], P [213.4 µg/mL of PO_4_^−^ equivalents], and K [169 µg/mL of K^+^ equivalents] solubilization). These strains are among the top performers for some of the screening tests, but also showed mid performance for some other skills.

Among them, five are able to solubilize phosphate, and two are able to solubilize potassium. In addition, the strain *P. nitroguajacolicus* AI is able to prepare both; however, was only isolated from unburnt places. We decided this because we consider these nutrients among the easiest to lose after a fire event. Finally, it is important to remark that four of the strains show three or more skills, making the consortium more coherent and consistent. At a species level, most of them are only present in unburnt places (some are residually in burnt locations), and were isolated from root or top soil samples. To better reference the abilities of these selected strains, we carried out quantification tests on each outstanding ability.

### 3.5. Microcosms Evaluation

#### 3.5.1. Soil Evaluation: EC, pH and Slaking Index Coefficient (SIC)

The microcosms were prepared simultaneously, and samples were taken for initial characterization (t = 0). The local microbiota of the soil was assessed at this point, obtaining a marginal population (1.57 × 10^2^ ± 0.62 CFUs/mg). A visual and parametric evaluation was performed during the 40 days of experimentation (Appendix A). Thus, the measure of electroconductivity (EC) was shown to be stable and minimal, oscillating between 0.0–0.02 during all the experiment. On the other hand, the pH measures (Figure 6a) showed how initially the microcosms supplemented with ashes were more basic (around 8.8) with respect to the control microcosms, which in general were shown to be more acidic (5.6). In the control microcosms (unburnt conditions, UB), the pH values oscillated from 5.3–5.9 during testing, where the microcosms with the consortium treatment were closer or above pH = 6 at some points. In the case of the burnt-mimicking microcosms (B), the general tendency was towards acidification with time (pH = 8.3); however, this tendency was more remarkable in those treated with the consortium (pH = 7.3–7.6), especially in the condition joining plants and consortium.

In the case of the aggregation, the slaking index coefficient (SIC) was measured during the experiment (Figure 6b). In the case of the burnt-mimicking microcosms, they initially showed slightly less aggregation (about 7.3) than the control microcosms (6.8). The values recorded were stable for the mock-treated microcosms in both conditions, meanwhile the rest of the treatments were gaining aggregation during the experiment. In the case of control microcosms (UB), the single treatment with plants or consortium showed similar results by the end of the assay (6.3), gaining about 0.4 points of aggregation, however, the combination of both treatments caused an increase in aggregation of about 1 point (5.8). Similarly, in burnt-mimicking microcosms (B), the single plant or consortium treatment caused an increase in aggregation of about 0.6 and 0.5, respectively. Moreover, the microcosms treated with both showed a SIC of about 6, which supposes an aggregation increase of 1.3 points.

#### 3.5.2. Microscopy and Granulometry Evaluation of Soil Aggregation Degree

As a first step, we described how the SIC descended in those microcosms where plant, consortium, or both were applied. Here we decided to evaluate visually the aggregation degree at the microscopy level (Figure 7). The pictures of both mock sets showed the complete disaggregation that their soils had at the end of the experiment. Moreover, in the case of burnt-mimicking microcosms, the presence of ashes reduced the soil weft in particle size, not showing any aggregation pattern due to the incorporation of the ashes. In the microcosms that only included seeds of *B. bituminosa*, they commonly showed a small chunk-like aggregation pattern, joining big-sized particles of the soil. Although this was clearer around the roots, this pattern was present along most of the superficial layer of soil. Focusing on the burnt-mimicking sets, the aggregates joined the ashes in a cement-like structure. As observed in both conditions, the plants were able to spread hairy roots along the main and secondary roots, which provided a filamentous matrix for the surrounding particles, but the adherence was not very intense since very few remained attached when the soil was sampled. Following this, the sets of microcosms treated only with the bacterial consortium showed a different aggregation pattern where different size particles appeared glued more commonly along the sample. In contrast with the effect of ashes and roots, here, in both cases, the weft did not look like a cement aggregation, but in a chunkier way. Finally, in the sets treated with plants and the consortium, the aggregation was similar to the previous patterns, with maybe a more compact structure in the case of the burnt-mimicking samples. In addition, for these samples, in general, hairy roots were able to capture more particles more firmly.

On the other hand, a granulometry assay was performed at the end of the experiment. Here both mock sets showed an identical distribution of size particles as the soil samples prepared at the beginning of the test, so no significant evolution was detected despite the condition applied. About the treated sets, beginning with the control microcosms, they all showed a clear gain in the >1 mm fraction. Here, compared to the mock set, the samples from the microcosms treated with plant, consortium, or both, showed and increment from about 5.5% to 21, 14, and 25%, respectively. The fraction of 1–0.45 mm significantly decreased from about 32% up to 27, 29, and 21%, respectively. Moreover, the 0.45–0.25 mm fraction was also decreased in the sets with only plants (from 35% to 28%). Despite the 0.25–0.063 mm fraction also slightly decreasing in all conditions, the <0.063 mm fraction, very relevant as an aggregation indicator, decreased in all the treatments, but was especially remarkable in the microcosms with plants and the bacterial consortium (from 6% to 3%).

Similar to the control microcosms, in the burnt-mimicking ones the >1 mm fraction increased notably in all treatments with respect to the mock (from 10% to 20, 25, and 27%). The fractions of 1–0.45 mm, 0.45–0.25 mm, and 0.25–0.063 mm were all slightly reduced, especially in the microcosms treated with the consortium (between 3–5%). On the other hand, in the <0.063 mm fraction we observed that the microcosms were three times less represented in the microcosms treated with only plants or only the consortium (from 5.5% in the mock microcosms to around 1.65%); however, the most remarkable reduction in this fraction was detected in the microcosms treated with both plants and the consortium combined, decreasing this fraction to less than 1% (five times less). This result is the most intense reduction in any of the fractions, and in any of the microcosms designed for this experiment.

#### 3.5.3. Main Nutrient Determination (N, P, and K)

To evaluate the final effect of the treatment on the main nutrients, we quantified the amount of soluble nitrogen (N), phosphorous (P), and potassium (P) 40 days after the beginning of the experiment. The final NPK values detected in the mock sets did not significantly change from the original amount, neither in the control (15, 25, and 54, respectively) nor in the burnt-mimicking (about 23, 34 and 68 mg/Kg, respectively) microcosm sets. However, all the treatments applied caused a change in the solubility of the nutrient, causing an increment in the detection of all the nutrients (Figure 8). Regarding the N, in the control sets, we observed that the treatment with plants increased the availability by almost three times, meanwhile, the set treated with consortium had a 50% higher amount of soluble N. In the burnt-mimicking sets, the increase was slightly lower (30–60%) than in the control sets, but in the sets with plants and in the ones treated with the consortium it was about 2.5 time higher. For P and K, the results showed similar patterns by treatment. Hence, the sets with plants increased by more than two times the soluble P and K, both in the control microcosms. In the burnt-mimicking sets, these values only rose when plants were combined with the consortium. The impact of the consortium treatment alone was an increase in the values by about 30–40% in all conditions.

#### 3.5.4. Germination Rate

The germination rate of the *B. bituminosa* seeds in the microcosms was evaluated during the 40 days (Figure 6c). Here, we included an in vitro germination assay as an optimal condition control. The treatment with the different bacteria, as well as with the consortium, showed a similar or slightly lower germination ratio compared to the untreated (data not shown). Considering the microcosms including seeds, the control sets showed almost double the germination rate compared to the rate registered in the burnt-mimicking sets. Focusing on the microcosms treated with the consortium of bacteria, in both conditions, the germination rate increased by 20 and 40% in the control and burnt-mimicking microcosms, respectively. Moreover, in the control microcosms treated with the consortium the germination rate was very similar to the one obtained under in vitro conditions.

#### 3.5.5. Plant Phenotyping

To evaluate the phenotype of the germinated seeds of *B. bituminosa* (Figure 9 and Appendix A), we used a statistical approach considering the different number of individuals obtained in each set. The evaluation consisted of the measure of the root, shoot, and biomass of the seedlings at the end of the experiment (40 days). Hence, the root size was similar between the sets treated with the consortium (about 9.5 cm); however, the roots of the seedlings growing in the burnt-mimicking sets without the consortium were significantly smaller (less than half). In the case of the shoot height raised by the seedlings, we found significant differences. In the sets of microcosms without the consortium and consortium-treated, the size of the seedlings in the burnt-mimicking microcosms were about 43% and 47% smaller than in the control set, respectively. Finally, the biomass of the seedling was evaluated by considering the full-plant dry weight. Here, again, the seedlings in the burnt-mimicking microcosms showed significantly less value. In the case of the sets without the consortium, the seedlings weighed 43% less, and in the case of the sets treated with the consortium, this reduction was up to 33%. When we consider the difference between treated and untreated sets in the burnt-mimicking sets, they showed a 46%, 35%, and 40% increase in root length, shoot height, and dry weight, respectively.

## 4. Discussion

In the context of the increasing magnitude of wildfires, the case of Los Guájares represents a model case for the Mediterranean area as one of the most common types of forests that were affected (wild and repopulated pine forests and Mediterranean scrubland). This area fits into a very exposed climatic region (coastal-Mediterranean) and occurs in one of the most common soils in the region (calcic cambisol). Sampling in the area was carried out four months after the fire, and we were able to verify homogeneity in the characteristics of the sampled soils. Thus, we assessed the accumulation of certain nutrients and ions in the burned soils (P, K, Mg, Ca, and Fe), all of which are easily washed away or lost due to erosion [6,14,19]. It is necessary to highlight that sample 3 is in the steepest location, so could be more subject to washdown after a fire event, causing some of the unexpected nutrient’s balance between burnt and unburnt samples (specially in P and K). In addition, in all the soils, we detected the emergence of *Bituminaria bituminosa* seedlings, which in turn began to emerge in the nearby unburned areas. Furthermore, this species is a legume capable of fixing nitrogen through interactions with rhizobacteria, even under rigorous conditions and droughts in the Mediterranean climate [84,85]. This gave us the possibility of using this species as a good model for studying soil recovery and essential nutrient fixation.

However, apart from compromising their viability, seed germination after fire can be conditioned by the pH, structure, or humidity of the soil. In our experiments, we were able to verify how the conditions of the burnt-mimicking microcosms substantially reduced seed germination [84,85]. Here, after combined application with the bacterial consortium, germination and plant growth improved. Even though the in vitro germination tests with each of the consortium strains did not show an evident improvement in the germination rate (data not shown), the combined effect of these bacteria caused an improvement in the soil structure, which may better retain the moisture, as well as a reduction in the pH of the soil, which could explain the improvement in germination. In addition, despite not having much information about plant growth-promotion in these species, some of the strains included in the consortium, such as *Pseudomonas koreensis*, *Paenibacillus lautus*, and *Paenarthrobacter nitroguajacolicus*, have been described as growth promoters in other legumes, which could explain the improvement in development in both microcosm conditions [86,87,88].

Considering the selection of bacterial strains as candidates for the consortium, we decided to perform a selection based on the two most relevant mechanisms for the recovery treatment: fixation of nutrients and aggregation of affected soils [89]. Therefore, we also decided to characterize the cultivable populations, which would allow us to distinguish the need to recover local populations in the affected soils or if it is sufficient with the emerging strains. Recent studies of the same type of soil and climate, as well as other more general ones, have shown that the populations that emerge weeks after fire events tend to be less diverse and less resilient, usually recovering their original populations over time as long as the ground does not erode and is lost. Rai et al. showed that in a short-term prescribed fire, such as low-intensity fire events, the recovery of the populations occurs rapidly [90]. However, higher intensity levels require a time frame that could be too late for soil recovery. Soria et al. also focused on the Mediterranean region, and this initial loss of general diversity is reflected, but an increase in some genera such as *Pseudomonas* has been reported, which is also observed in our studies [91]. However, our work, although limited to the cultivable portion of the population, found that the predominant genera in these soils were *Peribacillus* or *Bacillus*, as is also reflected in the studies by Matos et al. [92]. A recent study by Hrenović et al. also showed these trends as well as an increase in phosphate binders in emerging communities [93]. In our study, we were able to verify that both this group and the potassium fixators were present; however, they had a smaller population and were less effective than the strains isolated from the sampling points not affected by fire. Likewise, strains capable of oxidizing sulfur or producing siderophores to fix metal ions were underrepresented in the locations affected by the fire. This trend was also seen in some of the structuring abilities and improvements of plant development, since the number of isolates capable of producing biofilms and auxins was almost double in soils that were not affected by fire.

Moreover, we evaluated the nutrient availability under the different treatments. We contrasted in our study that the ashes increased the nutrient content on burnt-mimicking sets; however, the factor that substantially increased the nutrients, making N, P, and K more soluble, was the inclusion of plants. This was previously referenced in many works, as in Carvalhais et al., pointing to the release with the exudates of different organic acids [44,46,94,95,96]. Although a single application of the consortium did not show the same impact, the bacterial treatment also enhanced the presence of these nutrients, as frequently reported in this field [44,97,98,99]. More interesting was to validate the theory of many authors, and our own premise, regarding the combined use of bacteria and the plant effect in nutrients availability [44,45,100,101,102]. In our case, this combination showed the highest increase in all evaluated nutrients, which was especially evident in the burnt-mimicking sets.

In our particular case, it seems that despite having diverse populations, the strains with the necessary abilities to recover the soil are not as common as they should be, and we decided that our strategy needed to supply strains from unburnt soils. This motivated us to prepare our consortium with more strains isolated from locations not affected by fire. By studying cultivable populations and their capacities, we prepared a consortium or synthetic community (SynCom) treatment for our microcosms. To the best of our knowledge, this is the first attempt to treat the effects of fire on soil and accelerate its recovery with SynCom. Although the most convenient method could have been the individual treatment by strain, we considered that a small consortium in the synthetic community made more sense to cover most of the required skills, which no single strain seemed capable of covering. This approach has provided more promising results in complex systems, such as our object of study, as we can find in the studies by Schmitz et al., Yin et al., and Flores-Duarte et al. [78,103,104]. Apart from the improvements described before in the germination and development of the *B. bituminosa* seedlings, this treatment showed that it was able to improve the degree of aggregation, something that we verified both through the Slaking index, as well as by granulometry (Figure 6 and Figure 7). The microscopy visualization showed that this aggregation differed between the only plant and only bacterial consortium applications. Whereas the plants aggregated locally, deeper, and with larger particles, the consortium aggregated more superficially and with finer particles at a general level [56,105,106,107]. Despite these encouraging results, we were able to verify that the combined application of both treatments substantially improved these parameters. In addition, the pictures revealed that the consortium improved the number of hairy roots in *B. bituminosa* seedlings, which led to the anchoring and aggregation of larger particles, as was described for other environments and for other stressing conditions [108,109,110,111].

Hence, this approach may provide an effective treatment that accelerates soil stabilization and helps in ecosystem recovery. As some authors are claiming, a better knowledge of microbial roles after fire events, as well as their use as a treatment, may help create a relevant step forward in our strategies for soil recovery [9]. The next steps may include refinement of the consortium, co-formulation of emerging plants with bacteria, biosafety use of the treatments, and long-term follow-up, as other microbial application in nature require to succeed [28,112,113,114,115,116]. Although the best management of fires is the prevention and care of forests and bushes, our study shows that knowledge of local populations after fire events can provide effective biotechnological tools to offset the effects of fires on the soil and ecosystems.

## 5. Conclusions

The soils affected by intense fire events suffer from a serious imbalance in diversity and nutrients, conditioning the prospects of the affected ecosystems. This present work sought to understand the changes in local microbial diversity and apply corrective measures to accelerate the first phases of soil recovery. In this study, we found that cultivable microbial populations in fire-affected soils lost a variety of bacteria involved in the fixation and accessibility of essential macro and micronutrients. Through microcosm tests, we observed that the application of these absent bacteria together with emergent plants helped accelerate soil recovery. This treatment improved the aggregation of soil particles and the amount of available nutrients (N, P, and K), while improving the germination and development of the model plant *Bituminaria bituminosa*. However, these results are preliminary, and we believe that a longer evaluation period is necessary to discern the effects in seedling development, the degree of aggregation in the long term, or even to include other external factors in the microcosms. Despite being at an early stage, our work highlights the relevance of communities that are lost due to the effect of fire for rapid recovery, as well as their beneficial role for local emergent plants to accelerate soil recovery. In this point, we also consider it necessary to implement a testing treatment under real conditions (in field) in order to enable the use of this technology in the future. The use of local resources, rapid implementation, and low impact could be complementary methods to stimulate emerging plant populations and provide microbial communities that have disappeared. Therefore, this study opens the door to future bacterial treatments to help in soil recovery and limit effects such as erosion after fires, allowing the ecosystem time to regenerate with greater guarantees.

## Figures and Tables

**Figure 1 biology-12-01093-f001:**
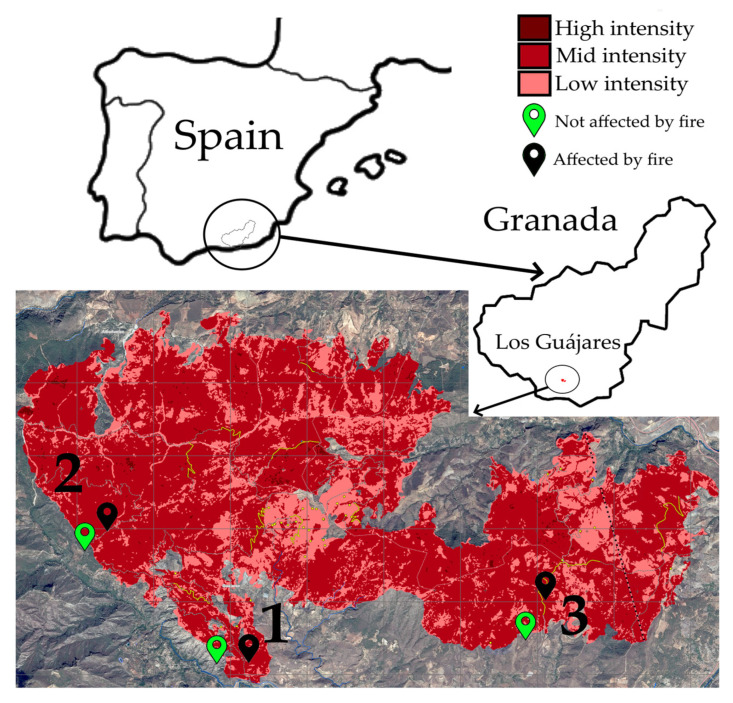
Sampling locations. The map infography shows the locations selected for sampling (the numbers 1, 2 and 3 refer to the sampling locations as listed in Table 1). Location pins in green indicate samples of soil collected in areas not affected by fire; black pins indicate samples of soil collected in areas affected by fire. The map of the total area affected by the Los Guájares forest fire is a modification of the one issued by the COPERNICUS Emergency Management Service website (https://emergency.copernicus.eu/mapping/ems-product-component/EMSR632_AOI01_GRA_PRODUCT_r1_RTP01/1, accessed on 10 May 2023), where the maroon color indicates the places where the intensity of the fire was high; red, where the intensity was intermediate; and pink, where the intensity was low. All the samples were obtained from locations where the intensity of the fire was intermediate.

**Figure 2 biology-12-01093-f002:**
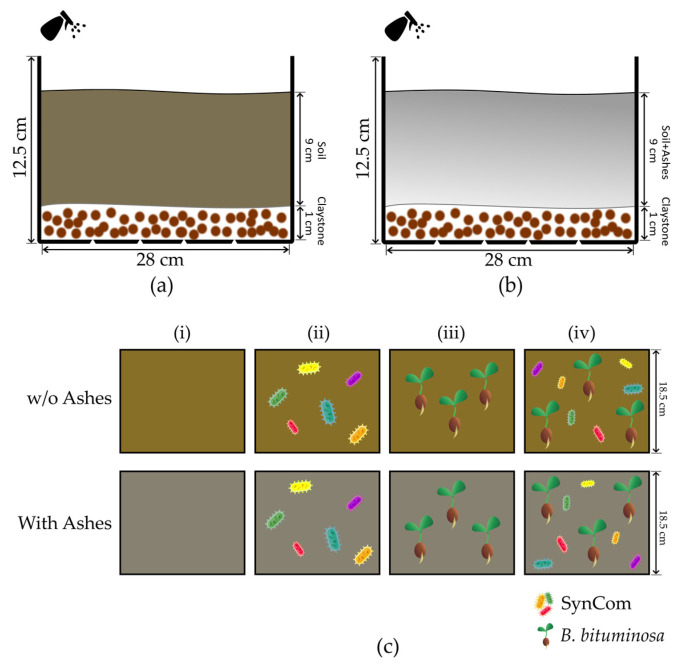
Proposed model and treatments for microcosms evaluation tests. The microcosms were prepared in 4 L volume in sandy soils (**a**) mixed with ashes in the case of the burnt evaluation model (**b**). In both conditions, relative humidity was maintained by regular water spraying, controlling the excess in case by using a draining system. Here, we established four treatments (**c**) for microcosms without (w/o) ashes and with ashes. Treatments included were (i) control soil, (ii) soil + synthetic bacterial community (SynCom), (iii) soil + *Bituminaria bituminosa*, and (iv) soil + SynCom + *B. bituminosa*.

**Figure 3 biology-12-01093-f003:**
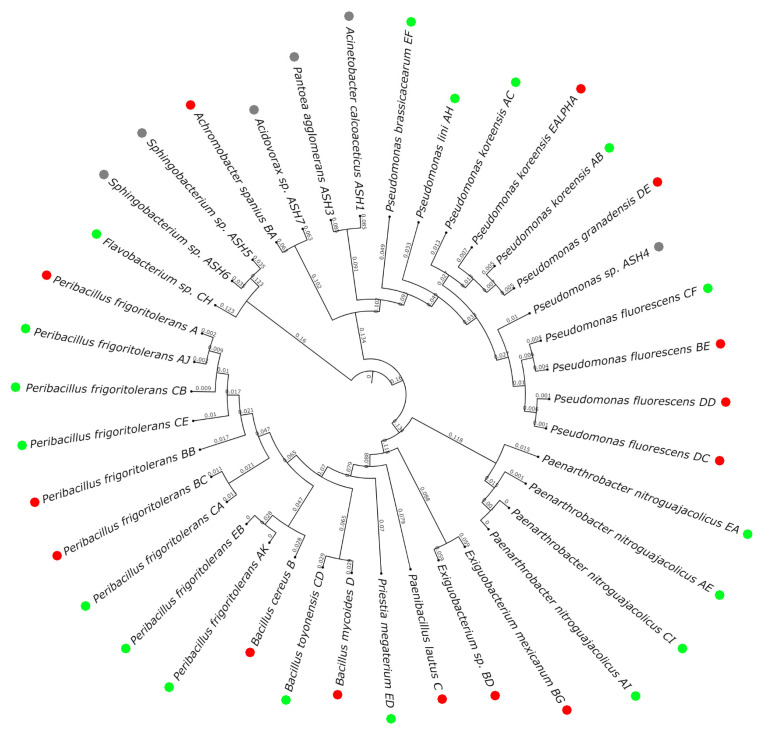
Population analysis. The circular phylogenetic tree shows the proximity of the strains isolated in this study, indicated by branch length. The strains labelled in green were isolated from unburnt locations; the ones labeled in red, from burnt locations; and the ones labelled in grey, from the mix of ashes.

**Figure 4 biology-12-01093-f004:**
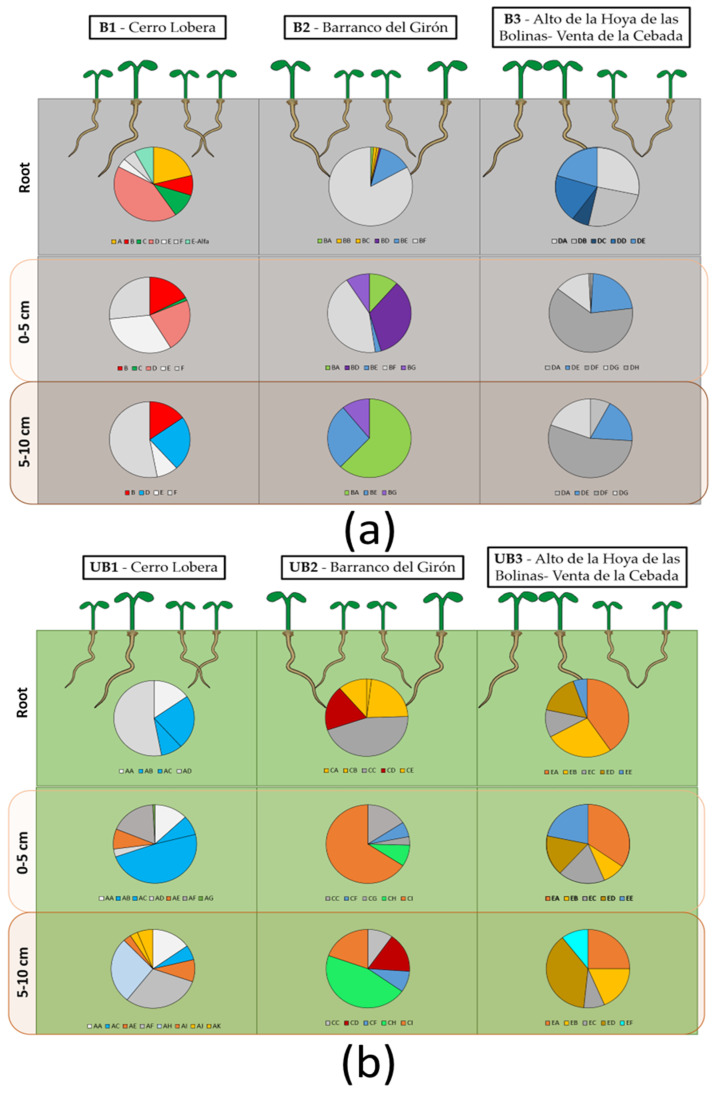
Distribution of culturable population in the different sampling locations. This infographic and pie charts show the distribution of the strains isolated in the three locations (1 for Cerro Lobera, 2 for Barranco del Girón, and 3 for Alto de la Hoya-Venta de la Cebada) for burnt ‘B’ (**a**) and their corresponding unburnt ‘UB’ areas (**b**). Here, the reddish-colored sectors represent the prevalence of *Bacillus strains*; the yellowish-colored ones, *Peribacillus strains*; the orangish-colored ones, *Paenarthrobacter strains*; the bluish-colored ones, *Pseudomonas strains*; the purplish-colored ones, *Exiguobacterium strains*; the goldish-colored ones, *Priestia strains*; and the greenish-colored ones gather *Achromobacter*, *Paenibacillus* and *Flavobacterium strains*.

**Figure 5 biology-12-01093-f005:**
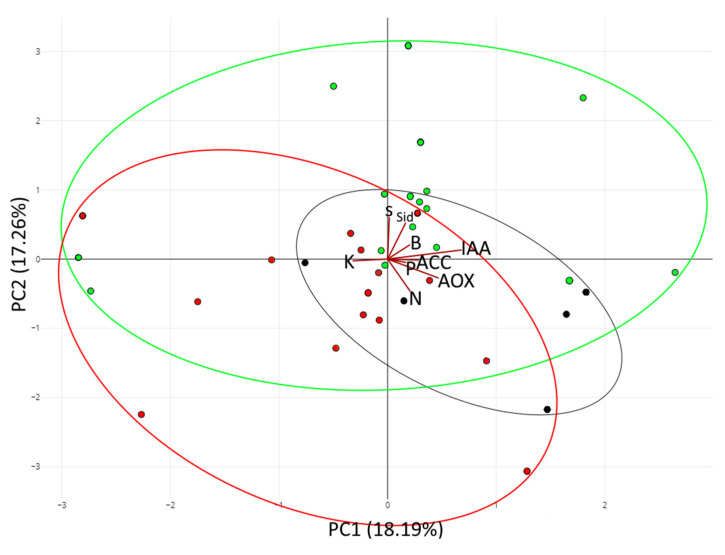
Principal Component Analysis (PCA). The first two principal components from a Principal Component Analysis using rlog transformed expression values. Principal Component Analysis (PCA) of strains isolated in unburnt locations (green), in burnt locations (red), and in the mix of ashes (black). The first principal component (PC1, x-axis) explains 18.19% of the variation in the data while the second principal component (PC2, y-axis) increases total explained variation to 17.26%. A confidence ellipsis at 99% is drawn for each group. Each trait evaluated was represented with a red vector in the graph. ‘N’ corresponds to nitrogen fixation; ‘P’ to phosphorus solubilization; ‘K’ to potassium solubilization; ‘S’ to sulfur oxidizing; ‘Sid’ to siderophores production; ‘B’ to biofilm production; ‘IAA’ to auxin production; and ‘AOX’ to antioxidant production.

**Figure 6 biology-12-01093-f006:**
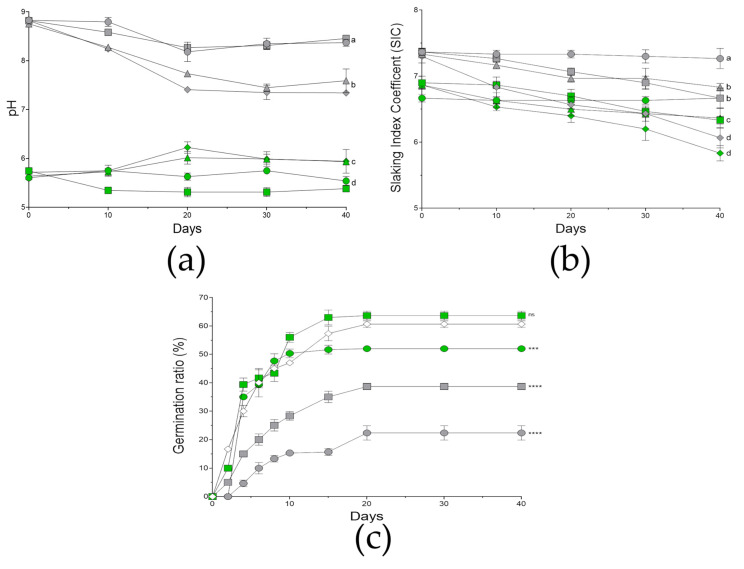
Evolution in microcosms. The line graphs show the evolution in the pH (**a**), the slaking index coefficient (SIC) (**b**), and the germination rate of *Bituminaria bituminosa* seeds (**c**) under each condition recorded in each microcosm (n = 3) during the 40 days of evaluation. Here, the circular-green markers stand for unburnt microcosms; the squared-green markers, for unburnt microcosms treated with the consortium; the circular-grey markers, for unburnt microcosms; the squared-grey markers, for unburnt microcosms treated with the consortium. The rhomboid-empty markers stand for in vitro germination rate recorded in parallel as control. The sets of data were compared using a two-ways ANOVA, where the letters indicate same significance level; alternatively, the asterisks represent a statistically significant difference at *p <* 0.001 ***, and *p <* 0.0001, ****; meanwhile, ^ns^ stands for groups with no statistical difference with respect to the in vitro control for germination. Error bars represent s.d.

**Figure 7 biology-12-01093-f007:**
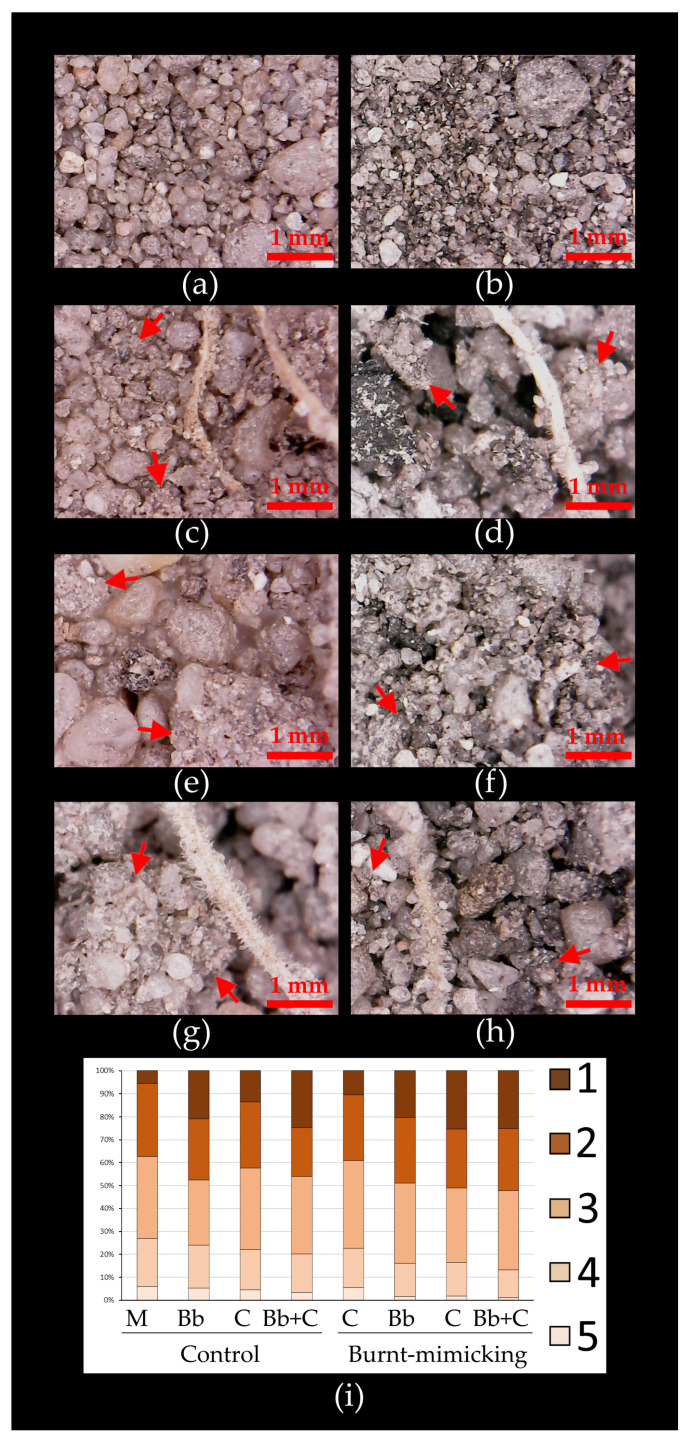
Aggregation induction by treatments in microcosms. The microscope pictures show the aggregation patterns of the control and burnt-mimicking microcosms for mock (**a**,**b**), treated with plant (**c**,**d**), with the bacterial consortium (**e**,**f**), and with both (**g**,**h**). The scale bars represent 1 mm in the pictures, and the red arrows indicate some of the more representative aggregation patterns detected, although these measurements do not allow us to evaluate patterns for each condition and treatment. The stacked column chart represents the granulometry fractions in % (**i**) 40 days after the beginning of the microcosm experiment. Here, ‘M’ stands for mock microcosm; ‘Bb’ for microcosm with plants; ‘C’ for microcosm with consortium; and ‘Bb + C’ for microcosm with both treatments. For the granulometry, color box 1 stands for >1 mm fraction; 2 for 1–0.45 mm; 3 for 0.45–0.25 mm; 4 for 0.25–0.063 mm; and 5 for <0.063 mm fraction.

**Figure 8 biology-12-01093-f008:**
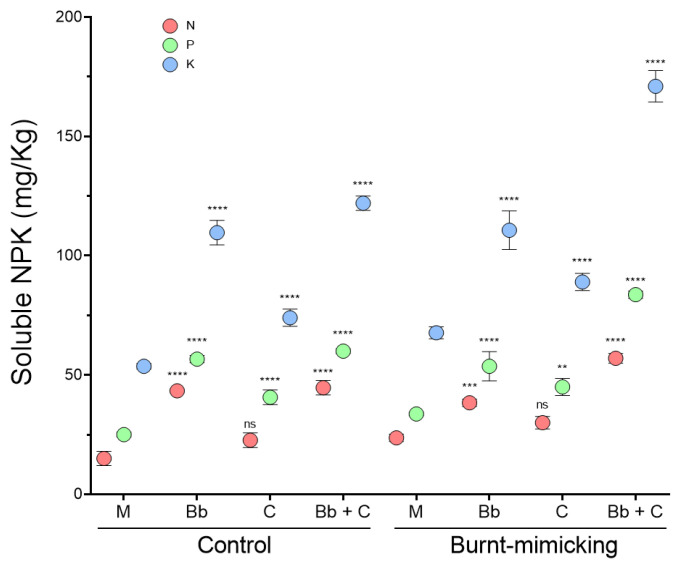
Determination of soluble nutrient content (N, P, K) in microcosms. The dot graph shows the amount (mg/Kg) of soluble nitrogen (N), phosphorus (P), and potassium (K) at the end of the experiment in each control and burnt-mimicking microcosm. The red dots stand for N; the green ones for P; and the blue ones for K. Here, ‘M’ stands for mock microcosm; ‘Bb’ for microcosm with plants; ‘C’ for microcosm with consortium; and ‘Bb + C’ for microcosm with both treatments. The sets of data were compared using a two-ways ANOVA. The asterisks represent a statistically significant difference at *p* < 0.01 **, *p* < 0.001 ***, and *p* < 0.0001, ****; meanwhile, ^ns^ stands for groups with no statistical difference with respect to each mock set as control. Error bars represent s.d.

**Figure 9 biology-12-01093-f009:**
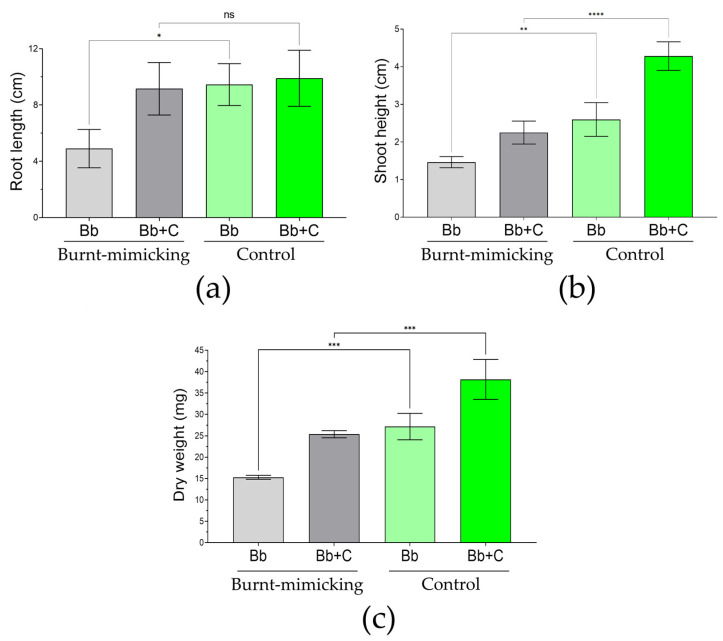
Phenotype evaluation of *Bituminaria bituminosa* plants germinated in the microcosms. The bar graphs show the root length (**a**), shoot length (**b**), and full-plant dry weight (**c**) in the plants germinated (n = 30) under burnt-mimicking conditions (grey-colored bars) and control conditions (green-colored bars) in each microcosm (n = 3) during the 40 days of evaluation. The sets of data were compared using a two-ways ANOVA, where the asterisks represent a statistically significant difference at *p <* 0.05 *, *p <* 0.01 **, *p <* 0.001 ***, and *p <* 0.0001, ****; meanwhile, ^ns^ stands for groups with no statistical difference with respect to the control. Error bars represent s.d.

**Table 1 biology-12-01093-t001:** Sampling locations. Samples were collected by pairs in close areas, which are defined by the corresponding number (for the map location see Figure 1). Each pair consisted of soils affected and unburnt by fire, and each one included 0–5 cm and 5–10 cm depth. GPS coordinates (World Geodetic System 1984, WGS84) and altitude (meters above mean sea level -mamsl-) are approximated.

No.	Name	Location	GPS	Altitude (mamsl)	Geological Contex
1	Burnt (B1)	Cerro Lobera	36°51′01.5″ N 3°36′25.4″ W	377	Marble with biotite ^1^
	Unburnt (UB1)		36°51′04.5″ N 3°36′40.1″ W	417	Marble with biotite ^1^
2	Burnt (B2)	Barranco del Girón	36°52′11.1″ N 3°38′12.9″ W	736	Marble with tremolite ^1^, squists
	Unburnt (UB2)		36°52′14.8″ N 3°38′35.1″ W	726	Quartz-squists and squists ^2^
3	Burnt (B3)	Alto de la Hoya de las Bolinas-Venta de la Cebada	36°51′50.5″ N 3°32′32.2″ W	745	Dolomites, limestones, marbles
	Unburnt (UB3)		36°51′27.4″ N 3°32′38.6″ W	731	Alluvial sediment, dolomites, marbles

^1^ Marble is a basic rock. Biotite is a phyllosilicate of iron and aluminium. It can also contain Mn, Ti, Na, Zn, and Mg. ^2^ Squists with staurolite, distena, anphiboles, and epidote, which mainly contain Al, Fe, Si, Mg, and Ca. Sources: Instituto Geológico y Minero de España (IGME), MAGNA scle 1:50,000; Atlas Nacional de España, Instituto Geográfico Nacional (based on the European Soil Data Centre (ESDAC)). European Commission, 2001.

**Table 2 biology-12-01093-t002:** Soil characterization. Samples collected from top and deep soils in burned and unburned areas were characterized in physical/chemical parameters, including texture, pH, electroconductivity (EC), organic matter (OM), as well as main macro- and micronutrients.

Sample ^a^	Deep (cm)	Field Texture	pH	EC ^b^	OM ^c^ (%)	Total N (%)	P (P_2_O_5_, mg/kg)	K (K_2_O, mg/kg)	Mg (mg/kg)	Ca (mg/kg)	Fe (mg/kg)
B1	0–5	Coarse	8.5	0.19	6.70 ^VH^	0.298 ^H^	478 ^VH^	434 ^VH^	947 ^VH^	4308 ^VH^	234 ^VH^
UB1	0–5	Coarse	8.5	0.01	1.05 ^L^	0.087 ^L^	<23 ^VL^	80 ^A^	168 ^VH^	2964 ^H^	55 ^H^
B2	0–5	Coarse	8.4	0.09	6.00 ^VH^	0.252 ^H^	446 ^VH^	799 ^VH^	1007 ^VH^	4972 ^VH^	659 ^VH^
UB2	0–5	Coarse	8.1	0.02	6.70 ^VH^	0.279 ^H^	24 ^VL^	165 ^H^	183 ^VH^	5323 ^VH^	108 ^VH^
B3	0–5	Coarse	8.2	0.06	3.80 ^H^	0.230 ^H^	530 ^VH^	412 ^VH^	479 ^VH^	5910 ^VH^	414 ^VH^
UB3	0–5	Coarse	8.1	0.03	5.80 ^VH^	0.227 ^H^	645 ^VH^	424 ^VH^	210 ^VH^	4701 ^VH^	141 ^VH^
B1	5–10	Coarse	8.3	0.03	3.90 ^H^	0.166 ^A^	53 ^A^	120 ^H^	508 ^VH^	3833 ^H^	116 ^VH^
UB1	5–10	Coarse	8.7	0.01	0.75 ^L^	0.035 ^VL^	<23 ^VL^	73 ^A^	186 ^VH^	3036 ^H^	42 ^H^
B2	5–10	Coarse	8.3	0.06	8.00 ^VH^	0.295 ^H^	50 ^L^	353 ^VH^	930 ^VH^	3763 ^H^	412 ^VH^
UB2	5–10	Coarse	8.1	0.01	4.00 ^H^	0.189 ^A^	24 ^VL^	130 ^H^	143 ^VH^	4069 ^VH^	91 ^VH^
B3	5–10	Coarse	8.2	0.04	2.90 ^A^	0.105 ^A^	63 ^A^	155 ^H^	212 ^VH^	3589 ^VH^	142 ^VH^
UB3	5–10	Coarse	8.1	0.01	2.20 ^A^	0.126 ^A^	190 ^H^	187 ^H^	116 ^H^	1675 ^A^	70 ^H^

^a^, ‘B’ stands for burnt areas, and ‘UB’, for unburnt ones; numbers indicate the location; ^b^, ‘EC’ stands for Electric Conductivity measurement; ^c^, ‘OM’ stands for Organic Matter; Fertility classes: ^VL^, very low; ^L^, low; ^A^, average; ^H^, high; ^VH^, very high.

**Table 3 biology-12-01093-t003:** Strains collection. This table shows the strains initially isolated in the three locations covered by this study. They are divided considering whether the location source was affected (burnt) or not (unburnt) by the wildfire. Unfortunately, it was impossible to replicate some of the strains after their identification (‘Unable to grow’), and some others were impossible to identify (‘Unidentified’) after several PCRs/sequencing attempts.

Location 1—Cerro Lobera			
Burnt		Unburnt	
Codename	Scientific Name	Codename	Scientific Name
A	*Peribacillus frigoritolerans*	AA	Unidentified
B	*Bacillus cereus*	AB	*Pseudomonas koreensis*
C	*Paenibacillus lautus*	AC	*Pseudomonas koreensis*
D	*Bacillus mycoides*	AD	Unidentified
E	Unidentified	AE	*Paenarthrobacter nitroguajacolicus*
F	Unidentified	AG	Unable to grow
E-alfa (EALPHA)	*Pseudomonas koreensis*	AF	Unidentified
		AH	*Pseudomonas lini*
		AI	*Paenarthrobacter nitroguajacolicus*
		AJ	*Peribacillus frigoritolerans*
		AK	*Peribacillus frigoritolerans*
Location 2—Barranco del Girón			
Burnt		Unburnt	
Codename	Scientific Name	Codename	Scientific Name
BA	*Achromobacter spanius*	CA	*Peribacillus frigoritolerans/simplex*
BB	*Peribacillus frigoritolerans*	CB	*Peribacillus frigoritolerans/simplex*
BC	*Peribacillus frigoritolerans*	CC	Unidentified
BD	*Exiguobacterium sp.*	CD	*Bacillus toyonensis*
BE	*Pseudomonas fluorescens*	CE	*Peribacillus frigoritolerans*
BF	Unidentified	CF	*Pseudomonas fluorescens*
BG	*Exiguobacterium mexicanum*	CG	Unidentified
		CH	*Flavobacterium sp.*
		CI	*Paenarthrobacter nitroguajacolicus*
Location 3—Alto de la Hoya de las Bolinas—Venta de la Cebada			
Burnt		Unburnt	
Codename	Scientific Name	Codename	Scientific Name
DA	Unidentified	EA	*Paenarthrobacter nitroguajacolicus*
DB	Unidentified	EB	*Peribacillus frigoritolerans*
DC	*Pseudomonas fluorescens*	EC	Unidentified
DD	*Pseudomonas fluorescens*	ED	*Priestia megaterium*
DE	*Pseudomonas granadensis*	EE	Unidentified
DF	Unable to grow	EF	*Pseudomonas brassicacearum*
DG	Unable to grow		
DH	Unable to grow		

**Table 4 biology-12-01093-t004:** Strains isolated from ashes. The table shows a collection of strains isolated from a mix of ashes (very low culturable population) as potential candidates for treatments. Unfortunately, it was impossible to identify some of the strains (‘Unidentified’) after several PCRs/sequencing attempts.

Mix of Ashes			
Codename	Scientific Name	Codename	Scientific Name
Ashes1 (ASH1)	*Acinetobacter calcoaceticus*	Ashes5 (ASH5)	*Sphingobacterium* sp.
Ashes2 (ASH2)	Unidentified	Ashes6 (ASH6)	*Sphingobacterium* sp.
Ashes3 (ASH3)	*Pantoea agglomerans*	Ashes7 (ASH7)	*Acidovorax* sp.
Ashes4 (ASH4)	*Pseudomonas* sp.	Ashes8 (ASH8)	Unidentified

## Data Availability

Not applicable.

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
