# Peer review of "Combined Use of a Bacterial Consortium and Early-Colonizing Plants as a Treatment for Soil Recovery after Fire: A Model Based on Los Guájares (Granada, Spain) Wildfire"

_biology, 2023, doi:10.3390/biology12081093_

Round 1
Reviewer 1 Report
I read the manuscript with high interest. This work is very actual now, under the global warming and forest fire increasing. The manuscript is well organized and nicely written. The presented results are very important and interesting to readers. The manuscript is well illustrated and very clear.
I would like to thank you for making an important contribution to investigating such a relevant topic.
However, there are some issues to address to, before the manuscript could be accepted.
1. Introduction part: could you, please, add a working hypothesis?
2. Materials and methods: how far were burnt and unburnt locations from each other?
3. Materials and methods: could you, please, add characteristics of climate, soils, vegetation in the studied region?
4. Lines 410-415 – what do you think about pH values, that were quite similar between burnt and unburnt samples?
5. Lines 418-420: according to table 2, there is an increase of OM content in burnt samples, why?
6. Lines 427-430: how do you think, why samples from point 3 are exception?
7. Lines 436-438: what is a reason of Fe-enrichment in burned soils?
8. Lines 442-444: what is a classification of substances level content? Need a reference.
9. Tables 3 and 4: first column has errors
10. Line 501: what is (a)?
11. Lines 510-513: why do you think burnt samples contained about 3.5 times more colony forming units (CFUs) per milligram of soil dry weight?
12. Lines 515-517: the same question as above
13. Line 550: need bracket at the end of sentence.
14. Lines 620-661: need to graphically display the results
15. Line 646: please, correct to: measurements
16. Figure 7 (i): Please, make a legend more readable
17. Figure 8: what three points above dots mean?
18. Lined 874-899: I suppose that this paragraph should be in introduction part.
19. SFig.1 (a): please, add units.
20. SFig. 2: please, add units. I suppose, that the sizes of sectors should be different, shouldn’t it?
Author Response
Firstly, I would like to show my appreciation to the reviewer for the time and dedication. Here we have tried to address all the comments and suggestions included. Hope you can find it complete and satisfactory:
- Introduction part: could you, please, add a working hypothesis?
Answer: Thanks for the comment! We have included our main hypothesis to make the point clearer in the introduction.
- Materials and methods: how far were burnt and unburnt locations from each other?
Answer: Very interesting. We considered the burnt places as really affected (not in a border), and the unburnt locations, as the closer, clearly unaffected. In any case, we determined not to change the biotope or landscape unit (same soil, vegetation and climate) Here, the maximal distance was in the point 3, where it was about 650 m. In location 1 and 2, these distances were of 340 and 175 m, respectively. A note about these distances was included in the main text. Thanks!
- Materials and methods: could you, please, add characteristics of climate, soils, vegetation in the studied region?
Answer: Thanks for the comment! The soil type is included in table 1, based on official mapping services of Spain. Moreover, we assessed in the field. Here we have included a note about climate and main vegetation units in the sampling area. We firmly believe this info is very useful, so we want to thank for this comment again.
- Lines 410-415 – what do you think about pH values, that were quite similar between burnt and unburnt samples?
Answer: Nice comment. Yes, the pH didn’t change substantially. We assessed the pH in the field with a probe, and then in lab, but the results were very similar. We have checked several studies and found that some have big changes in pH, but some other not, as in our case. Seems this is due to the initial pH of the soil and the buffer compounds that could be liberated during the fire event. For our microcosms, we considered more interesting to cause an initial difference in order to check if pH regulation may play a role. However, the case of the study seems to be not having that change due to the original pH and materials, not an exception, but a case already reported by other authors. Hope this is enough.
- Lines 418-420: according to table 2, there is an increase of OM content in burnt samples, why?
Answer: The sampled areas were sampled about half a year after the events (when we first get the authorization to do this, due to safety reasons), where the plants were already emerging. This used to cause a change in humic acids and exopolysaccharides. Main changes are in sample 1 and 2 in surface and 2 in deep. These samples are coincident with the higher number of emerging plants, as well the most contrast with the unburnt compared areas, where the soil was more mineral, so any input is very noticeable. However, this phenomenon is very particular, and we have not a better explanation, so we include a note about this in the main text.
- Lines 427-430: how do you think, why samples from point 3 are exception?
Answer: This is a great point, and we forget to address it, so we really appreciate this comment. Location 3 was the sample with the steepest profile. Moreover, some months in the region are enough to cause a substantial washing (this region from October to December is when rain the most and wind is more intense). Not all the elements characterized were in the same line, but P and K have been reported as some of the easier to lost. We have included some notes to clarify this point during the discussion section.
- Lines 436-438: what is a reason of Fe-enrichment in burned soils?
Answer: Thanks for the question. It’s reported that most of the metallic ions used to be increased in soils after fire. Here, chelation is lost and Fe-rich organic structures are mineralized, liberation of these and other ions in a mineral form. Normally this is more frequent in low pH soil, however used to be regular in most of the type of soils (as in NASA Earth Observatory (2020) or the citation below). The tendency with Mg or Ca seems in the same line, showing this is maybe our general case, accumulation of mineral forms after fire.
- Rust, A. J., Roberts, S., Eskelson, M., Randell, J., & Hogue, T. S. (2022). Forest fire mobilization and uptake of metals by biota temporarily exacerbates impacts of legacy mining. The Science of the total environment, 832, 155034. https://doi.org/10.1016/j.scitotenv.2022.155034
- Lines 442-444: what is a classification of substances level content? Need a reference.
Answer: Not pretty sure about this question meaning. Here, we just addressed a common characterization of soil components. We have not used a ‘classification of substances level content’. This is the national patter to evaluate soil composition (ref. INIAV: https://www.iniav.pt/solos-nutricao-vegetal-fertilizantes), that follows the European normative. If the meaning is about the fertility classes, this criterion is designed by INIAV as well, based again in the referential European patterns and normative for soil characterization. I’m sorry if this section caused some misunderstood, but it’s in the shape required by the organizations in the sector and requirements of soil quality control normative.
- Tables 3 and 4: first column has errors
Answer: With no more indications, it’s difficult to know which errors. However, we have introduced changes in some of the strain's code names to match with other figures, in case this is the problem. Moreover, we have eliminated the ‘ - ‘, that could bring some misinterpretation.
- Line 501: what is (a)?
Answer: Thanks for the comment! Just a mistake from previous version. Now is corrected in the revised version.
- Lines 510-513: why do you think burnt samples contained about 3.5 times more colony forming units (CFUs) per milligram of soil dry weight?
And 12. Lines 515-517: the same question as above
Answer: Thanks for the comment. Here many authors have reported similar numbers (in the discussion they are included), and used to be because of an increase in the availability of nutrients and the lower competence. This is a normal evolution after fire effect in which, during the next months, the microbes increase in affected soils in a peak that may drastically descend once the nutrients are washed away or the competence is increased (other phenomena may affect this). In brief, this kind of evolution is corresponding with the expected in r-type organisms or r-strategists, which explode the resources rapidly as they are present (fire survivors or new colonizers). Emerging populations use this strategy to colonize an empty ecological niche or space, but are displaced by K-strategist once the ecosystem recover.
- Line 550: need bracket at the end of sentence.
Answer: Done! Thanks!
- Lines 620-661: need to graphically display the results
Answer: Thanks for the comment. We really understand your recommendation, however, the amount of data creates huge graphs very difficult to edit in a paper-like format, creating many difficulties to visualize. Due to this, we decided to upload all this data to an accessible FigShare file. This includes the rankings, so it’s very easy to follow them. Moreover, we are following the indication of representativity of graphs marked by the editorial line of this journal. Hope you understand it and consider that we are evaluating the quality of the results here by text instead of a lot of meaningless graphs difficultly legible and interpretable due to the number of strains included in each test.
- Line 646: please, correct to: measurements
Answer: Done, thanks!
- Figure 7 (i): Please, make a legend more readable
Answer: Done, thanks! We have substituted the legend for a 1-5 code described in the figure legend.
- Figure 8: what three points above dots mean?
Answer: The vertical dots? They mean each nutrient detected, as it’s described in the figure legend: N, for nitrogen, P, for phosphorus and K, for potassium. We have included these notes in the figure legend to facilitate. Hope this solves the comment.
- Lined 874-899: I suppose that this paragraph should be in introduction part.
Answer: Yes, but we already include some of these concepts in the introduction. We just included here in this shape to help the reader to refresh the origin of the main ideas we’re treating in the discussion. We consider this helps to compile the main statements before address the topics to discuss. Hope this is ok.
- SFig.1 (a): please, add units.
Answer: Thanks! We have added now.
- SFig. 2: please, add units. I suppose, that the sizes of sectors should be different, shouldn’t it?
Answer: The units here are independent strains (number of strains), so we included in the legend of the figure. The Venn diagrams are accepted to be represented in balanced or representative bubbles, but it’s just a personal aesthetic decision, and we like more in this shape. Hope it's ok.
Reviewer 2 Report

The language must be improved.
Author Response
Firstly, I would like to show my appreciation to the reviewer for the time and dedication. Here we have changed many expressions and terms that were not helping to the text reading. However, hope you consider that this manuscript was read and cleaned by a native colleague, so we have written it under his supervision and advice. Hope the changes applied are improving the text. Thanks again!
Reviewer 3 Report
Dear Authors,
Your manuscript entitled "Proposal for a new bacterial treatment for soil recovery and improvement of the recolonization of emergent plants after forest fires, based on the case of Los Guájares (Granada, Spain)" has been reviewed,
This article deserves attention, since it highlights a very important topic from an environmental and microbiological point of view.
This paper is well written in English, its design and its methodology are very clear.
Kindly find below a list of my remarks (minor and major ones):
01- The title of this article is very long, I suggest to replace "based on the case of Los Guájares (Granada, Spain)" by "a case of Spain"
02- The manuscript is very long, Authors are invited to reduce some paragraphs.
03- In the Introduction section, Line 48, after "... in terms of burned or affected surfaces", Authors are invited to add the following article as reference:
Ref: Heavy Metals Toxicity and the Environment
04- In the Introduction section, Authors do not talk in details about Siderophores, they are invited to talk (to define) about Siderophores and there role in bacteria, and also about phytosiderophores and its role in plants. They can use the following references:
Ref: Siderophores in environmental research: roles and applications.
Ref: A Review of Pseudomonas aeruginosa Metallophores: Pyoverdine, Pyochelin and Pseudopaline
05- In the Introduction section, Line 94, When authors talked about iron and pH, they are invited to talk about Fe2+ and Fe3+. And then they can talk about iron chelation by siderophores. They can use this article as a reference for this point:
Ref: Chelating mechanisms of transition metals by bacterial metallophores “pseudopaline and staphylopine”: A quantum chemical assessment
06- In the whole manuscript, Authors highlights on just one heavy metal which is iron, but you know many other metals play an important role in soil, metals such as Nickel, Cobalt, Copper, etc... Therefore I would like to ask you, why you did not take these metals into consideration in the present study? Its presence in soil does not affect the quality of this last?
07- In the whole manuscript authors are invited to follow the rules regarding the name of bacterial species, the full name followed by its abbreviation between parenthesis, then they can use the abbreviation in the manuscript. Example: Pseudomonas fluorescens (P. fluorescens).
08- Same remark as the point number (07), for the name of "Bituminaria bituminosa"
09- In the Materials and Methods section, Lines 223-233, Authors are invited to replace "min" by "minutes" and "s" by "seconds".
10- In the Materials and Methods section, Lines 223-233, Authors are invited to indicate the size of amplified amplicons.
11- In the Materials and Methods section, Paragraph "Siderophores production", Authors are invited to indicate which Siderophores were produced??
12- In the Materials and Methods section, Paragraph "Biofilm production", Authors are invited to indicate which bacterial strains were used in the production of biofilms?
13- In the Results sections, Lines 436-438, Authors are invited to indicate the amount of iron (Fe) is of which ion? Fe2+, Fe3+ or Both?
14- Concerning the Figure 6, Authors are invited to add keys for the figures.
15- Concerning the Figure 7 (i), legends on the figure are not clear, Authors are invited to put them in a larger font.
Best Regards,
Author Response
Firstly, I would like to show my appreciation to the reviewer for the time and dedication. Here we have tried to address all the comments and suggestions included. Hope you can find it complete and satisfactory:
01- The title of this article is very long, I suggest replacing "based on the case of Los Guájares (Granada, Spain)" by "a case of Spain"
Answer: Thanks for the comment! We really understand your recommendation, and we also consider it is a long title, longer than we like as well. However, we consider it is important to point this is a localized case. We need to show that each fire can follow a different process, but our case may serve as an initial point. We have applied some changes to short the title, but this item is considered necessary in order to avoid concerning or situations in other cases to compare.
02- The manuscript is very long, Authors are invited to reduce some paragraphs.
Answer: We really appreciate this comment. We are aware of how long is the manuscript, and we were worried about it. We have tried to implement some paragraph shortening. Furthermore, we hope the reviewer can also consider that, due to the diversity of approaches, tests and results addressing, the text is difficultly able to be shorter. External colleagues have even pointed that some aspects are very rapidly addressed in the text, and asked us to implement more in these sections. We have tried to balance and even though the text is very long. As mentioned, we have shortened some parts, but we feel we cannot cut more without losing relevant information or analysis. Hope this is ok, thanks!
03- In the Introduction section, Line 48, after "... in terms of burned or affected surfaces", Authors are invited to add the following article as reference:
Ref: Heavy Metals Toxicity and the Environment
Answer: It’s very interesting and we have considered the citation. Thanks at all!
04- In the Introduction section, Authors do not talk in details about Siderophores, they are invited to talk (to define) about Siderophores and there role in bacteria, and also about phytosiderophores and its role in plants. They can use the following references:
Ref: Siderophores in environmental research: roles and applications.
Ref: A Review of Pseudomonas aeruginosa Metallophores: Pyoverdine, Pyochelin and Pseudopaline
Answer: This is a nice topic, and we have included some more info and citations as recommended (also in discussion). However, I feel this is not the main topic of the manuscript to dedicate more space of information here. Hope you understand that the work is dense enough, and we cannot expand it more only in this line. The comment is very valuable, and we really appreciate your inputs in this case!
05- In the Introduction section, Line 94, When authors talked about iron and pH, they are invited to talk about Fe2+ and Fe3+. And then they can talk about iron chelation by siderophores. They can use this article as a reference for this point:
Ref: Chelating mechanisms of transition metals by bacterial metallophores “pseudopaline and staphylopine”: A quantum chemical assessment
Answer: First, thanks at all, we realize we didn’t go enough over the Fe and siderophores in this paper, not enough at least. Here, we hope you understand for us, the production of siderophores is an extra trait to be evaluated. As well as we didn’t go deeper for nitrogen fixation paths, enzymes behind P or K solubilization or the king of compounds produced during biofilm formation, we don’t address the specificities of the siderophores. All the traits evaluated are included as a way to include most of the required skill for a soil recovery. Only positive or negative, and in case, the amount produced to select the best of the strains. Considering this, we feel that incise much more in siderophores and not the others, having no more relevance to anyone of them in this work, is not necessary. However, as mentioned above, we have included more references and literature you provided here on the topic because we consider it is interesting, and we didn’t address it correctly. We really appreciate your comment and hope now is ok.
06- In the whole manuscript, Authors highlights on just one heavy metal which is iron, but you know many other metals play an important role in soil, metals such as Nickel, Cobalt, Copper, etc... Therefore, I would like to ask you why you did not take these metals into consideration in the present study? Its presence in soil does not affect the quality of this last?
Answer: Again, a very interesting point. Our main concerning after fire is to evaluate if the main nutrients are washed away. Despite heavy metals could be interesting to be addressed, Fe is the only heavy metal that really have a big impact in plants growth, and not toxicity. However, we understand that after fire, some other heavy metals can be increased, causing toxicity, maybe. This is a very interesting topic, but out of the main line we worked in this manuscript. Our main point is the nutrients availability facilitated by microbes, and how to use them to help in the soil recovery. Finally, our soil studies are not including heavy metals as you proposed because they used to be minority in these environments and are not playing such a relevant role in these soils or their recovery, as our main topic is focused on. We really appreciate your comment to highlight this and hope this is ok.
07- In the whole manuscript authors are invited to follow the rules regarding the name of bacterial species, the full name followed by its abbreviation between parenthesis, then they can use the abbreviation in the manuscript. Example: Pseudomonas fluorescens (P. fluorescens).
Answer: Really appreciate! We have defined in this way, helping in saving space and shortening the paragraphs. Thanks!
08- Same remark as the point number (07), for the name of "Bituminaria bituminosa"
Answer: Done! Thanks!
09- In the Materials and Methods section, Lines 223-233, Authors are invited to replace "min" by "minutes" and "s" by "seconds".
Answer: Done! Thanks!
10- In the Materials and Methods section, Lines 223-233, Authors are invited to indicate the size of amplified amplicons.
Answer:
11- In the Materials and Methods section, Paragraph "Siderophores production", Authors are invited to indicate which Siderophores were produced??
Answer: In all the traits tested, we are only interested into know if the strains perform the trait or not, and in case yes, how much they do, in order to select the strains for the treatment. The type of siderophore produced could be interesting, but not definitive, as they are not the enzymes behind the nitrogen fixation, or the ones for P and K solubilization, or the composition of the biofilm produced, for this work. All this is not very relevant for the present study; however, we consider that deeper knowledge could be interesting for more detailed studies in the future, when accurate knowledge in this could be great to optimize the treatments. Hope this is ok.
12- In the Materials and Methods section, Paragraph "Biofilm production", Authors are invited to indicate which bacterial strains were used in the production of biofilms?
Answer: The biofilm production was assessed in all the strains isolated in the different sampling location, as the rest of the other tests. We indicated in an initial paragraph, but if it’s relevant we can include in all the paragraphs for each test, despite it could be repetitive. We used Pseudomonas putida as a reference in order to evaluate the biofilm production level of our strains. Hope this answer is ok.
13- In the Results sections, Lines 436-438, Authors are invited to indicate the amount of iron (Fe) is of which ion? Fe2+, Fe3+ or Both?
Answer: The iron was detected by using AAAc-EDTA (Lakanen) and Atomic Absorption flame Spectrophotometry (PE-016-LQARS/LAS) as INIAV proceeded, and European regulation ask for soil. These approaches calculate the Fe using ferrous ammonium sulfate as standard. I don’t have more information to provide, but the technician that prepared this characterization as service just mentioned this. So, based on this I guess is Fe (II), but only as standard, guess is not differentiating the Fe (II) or Fe (III) in the soil, but estimating total Fe amount. Hope you may find this ok.
14- Concerning the Figure 6, Authors are invited to add keys for the figures.
Answer: We fully understand the suggestion, as we have tried before. However, the result was more confusing, even adding only representative ones. We tried to show as clean as possible and this was the best option. We are trying to follow journal editing advices in order to show the figures clean and self-representative, and the addition of keys was adding very few values and a lot of mess. Hope this is ok. Thanks a lot for the comment!
15- Concerning the Figure 7 (i), legends on the figure are not clear, Authors are invited to put them in a larger font.
Answer: The units here are independent strains (number of strains), so we included in the legend of the figure. The Venn diagrams are accepted to be represented in balanced or representative bubbles, but it’s just a personal aesthetic decision, and we like more in this shape. Hope it's ok.
Round 2
Reviewer 1 Report
Authors have taken into account all the provided notes and comments, article could be accepted in the present form for publication.
Author Response
We really want to express our gratitude for the comments and suggestions!
Reviewer 2 Report
This work, “Proposal for a new bacterial treatment for soil recovery and improvement of the recolonization of emergent plants after forest fires, based on the case of Los Guájares (Granada, Spain)” of Niza Costa et. al,
is very interesting and the approach of soil restoration after fire with a bacterial consortium and the addition of plant species is innovative.
However, the work is not well organized and for this reason I recommend an extensive review.
It is not clear the purpose of the work, slowly that you read it you understand the intention of the authors and why they go to investigate certain aspects. But this should be very clear from the outset, and I understood it at the end of the debate by asking myself many questions every time I read analyses.
The period after the fire is short (4 months), is not a sufficient time to actually evaluate both the bacterial community that is definitely in a phase of dynamic succession, I would have evaluated at least two additional sampling times to see if it remained constant to continue the study (7 months and 1 year).
The arrangement of the microcosms from the beginning is not clear. From what type of soil where they filled? with that championship and therefore with that actually damaged by the fire? or with a standard ground? Why is it that you can talk about soil recovery, about undamaged soil? If that was used, I missed it.
Many parts of the work are redundant and dispersive not directed! In a scientific work I expect a greater focus.
Below are all my comments and specific comments.
I think the purpose of the work is not very clear. The authors, for the recovery of burnt soil, go to evaluate a bacterial consortium that they choose on the basis of bacteria isolated from burnt soils and not in common with the treatments as well, and which have metabolisms involved in the Sulphur and phosphorous cycle. Now, these are present in soils four months after the fire... why should they make the soil more fertile?
with a view to recovery I would probably not expect these bacteria in a subsequent sampling a year from now... and perhaps use different bacteria.
Plus, they set up microcosms to evaluate what? it's not clear... they don't say they will evaluate recovery through soil aggregation... but afterwards they do, and this, how should it affect recovery? they don't explain it.
In essence, I suggest that it should be made clear why they are doing these experiments and the criterion behind them. The introduction is very broad and deals with fires in general but it could be more focused on these aspects that I highlight earlier. What other work has been done like this? What is known?
I appreciate that the title of the article is descriptive but, in my opinion, it is too long. I would suggest removing "proposal" and rewording the title.
Simple Summary:
L.14 What is meant by soil fixation?
Abstract:
L.19-21 The introduction is very long and often going outside the topic of fires in Spain. I would focus more on the location where the fire occurred "Los Guájares (Granada, Spain)".
Introduction:
L.40 due to the increase
L46-48 please rephrase the sentence. Is not clear which fire (intentional or natural) is the object of sentence.
L75 remove: regarding nutrients ”One of the most… “
L.81 released by
L91 “..positive impact on soil micronutrients such as Mg and Ca carbonates”. “however, not in all fires was recovered an increase of soil ph; in fact,…”
L124 This sentence needs a reference you can see this paper “Santini et al Catena 214 (2022) 106234 “.
L160, 192 unburnt, please use always the same name (unaffected or unburnt).
Table 1: soil type, if is the same you can report directly in the main draft and delete from the table.
The different depths of the soils were treated as different samples?
Section 2.3, and from soil and ashes? Did you extract the DNA directly from the soil and ashes? and then use it for sequencing? how?
I would prefer there to be a distinction between the three different sections: soil, plant and ash and how the analyses were done on them. Also within the same section, but well divided and clearer.
2.5.1 'soil structuring skills': Biofilm production
I do not agree with calling this paragraph this because the ability of microorganisms to produce biofilm depends on their ability to secrete molecules intrinsic to the organism itself and this... does not determine soil structuring. Please remove.
2.6 In microcosms, is the inserted soil that was sampled directly on site? Or was it generated by a mixture of factors?
Figure 2. I like figure 2. only the two conditions are practically mirror-image except that in one of them there is ash. in section c in the first panel is w/o ash, does it stand for with or without? in which case the grey panel could be removed and specified that the number of treatments is 8 (n=8).
L.375-377 sentence very long… not clear.
L.379 remove During the test and start with “Every”
Figure 5. Please improve the quality of image and also increase the size of the letters (N, S…etc)
3.4 since you chose these specific strains for soil recovery, did you also evaluate how the population varied over time after inoculation? by CFU?
L.851-872 I'm confused, what does this part have to do here? shouldn't it go in the soil aggregation part? so 3.5.2. Please change.
L.874 “weather condition” Climate change.
L.877-878 “whose pop…” is redundant
L.887-889 The discussion is well written but it is too long, we must focus more on the discussion of the significant results rather on why this study was made, which, instead, should be written better in the final part of the introduction.
L955 “another aspect…” Remove because you don’t tell nothing about other aspect. In our study..
L.956 Once again, these explanations (thank you because now it is clearer the intent of the research) do not go in this section, but in the introductory part so as to read the work ready for what you expect (things you discover during the reading).
L.981 which micrographs?
L.990-1005 This part, instead, seems a conclusion and future prospects.
5. There is a redundancy of "in this study". Try to define what has been discovered more directly (maybe you can add here the future perspective – see above).
L.1022 “soil fixation” doesn’t mean nothing!!
English language can be improved.
Author Response
Dear reviewer,
Frist let me thank you for your comments and suggestions. They were very useful! Here we have tried our best to cope with them:
It is not clear the purpose of the work, slowly that you read it you understand the intention of the authors and why they go to investigate certain aspects.
A: Thanks! Yes, we have checked this during the text, so we have tried our best to improve this aspect.
The period after the fire is short (4 months), is not a sufficient time to actually evaluate both the bacterial community that is definitely in a phase of dynamic succession, I would have evaluated at least two additional sampling times to see if it remained constant to continue the study (7 months and 1 year).
A: Our aim was to help in soil stabilization as soon as possible because here, in the Mediterranean region, rain period used to be in autumn, right after the hottest period when the wildfires happen. Our idea was not that about define populations in general in each point, but help in the soil stabilization and recovery as soon as possible (normally first 2-3 months are dedicated to the urgent tasks of hill stabilization and recovery burnt material). Moreover, this is the timing when main emerging plant species were germinating in a generalized way, allowing us to select a common species and analyze in this context. Of course, a longer characterization would be very useful, hope to have this chance during next years. We really appreciate this comment and hope our answer is ok. We have also included some of these concepts in the main text.
The arrangement of the microcosms from the beginning is not clear. From what type of soil where they filled? with that championship and therefore with that actually damaged by the fire? or with a standard ground? Why is it that you can talk about soil recovery, about undamaged soil? If that was used, I missed it.
A: Thanks for the comment! As this could cause some misunderstanding, we have worked this part specifically in order to clarify all the questions along the text (particularly in the ‘Materials and Methods’ section). In brief, we decide to use the less aggregated kind of soil available (a sandy soil) in order to detect any increase in the aggregation cause by our treatments. Any change in this soil, way less aggregated that the natural case in our natural soil case, may suppose a promising option to consider for future treatment in natural soils. We have changed some sentences here in order to avoid confusion. ‘Recovery’ is only applied as a future perspective here, then, for our experiments we talk about improved soil aggregation rate, as a factor that may indicate the effect of our tests.
Many parts of the work are redundant and dispersive not directed! In a scientific work I expect a greater focus.
A: We really appreciate this comment. Sometimes we always read the same as a ‘tunnel effect’ and it’s difficult to detect this problem. We also faced the difficulty to cope with different perspectives to make it more complete, however, this is making difficult the coherence. Based on your comments, we have rearranged several parts of the text to make it clearer and more direct. Thanks!
I think the purpose of the work is not very clear. The authors, for the recovery of burnt soil, go to evaluate a bacterial consortium that they choose on the basis of bacteria isolated from burnt soils and not in common with the treatments as well, and which have metabolisms involved in the Sulphur and phosphorous cycle. Now, these are present in soils four months after the fire... why should they make the soil more fertile? with a view to recovery I would probably not expect these bacteria in a subsequent sampling a year from now... and perhaps use different bacteria.
Plus, they set up microcosms to evaluate what? it's not clear... they don't say they will evaluate recovery through soil aggregation... but afterwards they do, and this, how should it affect recovery? they don't explain it.
In essence, I suggest that it should be made clear why they are doing these experiments and the criterion behind them. The introduction is very broad and deals with fires in general but it could be more focused on these aspects that I highlight earlier. What other work has been done like this? What is known?
A: Thank you for your comment and general overview. I will try to address this in order, but consider that we have been performing changes to elucidate or clarify all these aspects along the text as well. Here, we compared the culturable populations of microbes in both the affected and unaffected areas. Through this, we can better understand which populations remain and which are lost. Moreover, the study of their nutrient-related skills is very important in order to avoid the usual leakage of nutrients occurring in Mediterranean ecosystems after fire due to the loss of soil structure (due to autumn rains that used to follow immediately after the summer fire season). Normally, these nutrients are in a mineral form after fire, which makes them inaccessible to plants, hindering their emergence and spreading. Microbes can play a role in changing the form of certain nutrients (such as P and K solubilization or sulfur oxidation) to make them more accessible and facilitate plant development. With the study of populations in affected/unaffected soils, and the screening for microbe skills, we can also elucidate how good the remaining microbes are in affected soils, and compare if they are enough to carry on these processes, or conversely, the part of the population that was lost shows better performance. After comparing, we detected both dramatic descends in the strains able to perform most of the skill, and the efficacy of their performance. Therefore, we decided to incorporate strains from unaffected soils as part of the treatment to accelerate the natural recovery ratio. We hope this explanation is acceptable.
I appreciate that the title of the article is descriptive but, in my opinion, it is too long. I would suggest removing "proposal" and rewording the title.
A: Thanks for the suggestion! We firmly believe the term ‘proposal’ helps to understand this is a new option that may help once developed in the future. However, we understand this may cause confusion, so we have applied changes in our title. Thanks!
Simple Summary:
L.14 What is meant by soil fixation?
A: Thanks for the comment! There are not consensual terms for this kind of phenomena, so we decided to use the most common term in Spanish. The translation may not mean the same or cause confusion, so we have decided, following your advice, to change it by ‘soil stabilization’ or ‘soil recovery’ depending on each particular case.
Abstract:
L.19-21 The introduction is very long and often going outside the topic of fires in Spain. I would focus more on the location where the fire occurred "Los Guájares (Granada, Spain)".
A: Done! Thanks for the suggestion! We have decided to include more info about Los Guájares environmental conditions later on in the text.
Introduction:
L.40 due to the increase
A: Done, thanks!
L46-48 please rephrase the sentence. Is not clear which fire (intentional or natural) is the object of sentence.
A: Done, thanks!
L75 remove: regarding nutrients ”One of the most… “
A: Done, thanks!
L.81 released by
A: Changed, thanks!
L91 “..positive impact on soil micronutrients such as Mg and Ca carbonates”. “however, not in all fires was recovered an increase of soil ph; in fact,…”
A: We have a addressed some changes in these phrases. Thanks!
L124 This sentence needs a reference you can see this paper “Santini et al Catena 214 (2022) 106234 “.
A: Great, really appreciate! Done!
L160, 192 unburnt, please use always the same name (unaffected or unburnt).
A: Sure, it’s confusing. We have homogenized this in the text. Thanks!
Table 1: soil type, if is the same you can report directly in the main draft and delete from the table.
A: Done, thanks!
The different depths of the soils were treated as different samples?
A: Yes, we detected that in first 5 cm were especially rich in ashes, but below this depth, the soil seemed less affected. Moreover, most of the sampling locations had no more than 9-12 cm before bedrock-like layer of soil, so we decided to use only this fraction (5-10 cm) for all of them. We added some notes to clarify this.
Section 2.3, and from soil and ashes? Did you extract the DNA directly from the soil and ashes? and then use it for sequencing? how?
A: We used samples from mix of ashes, and 2 different depths of soil. We didn’t extract DNA, we isolated culturable populations in LB (rich medium) plates. Once we isolated pure culture of the strains, we did DNA isolation of them and 16s rRNA PCR (fingerprinting). Same strains were collapsed in just one for the next coming tests. Notes about this were added to the text. Thanks!
I would prefer there to be a distinction between the three different sections: soil, plant and ash and how the analyses were done on them. Also within the same section, but well divided and clearer.
A: Thanks for the suggestion. We have applied changes to clarify this. We decided to describe together because the method is very similar and we want to avoid reiteration. However, we have separated their peculiarities to help in the interpretation.
2.5.1 'soil structuring skills': Biofilm production
I do not agree with calling this paragraph this because the ability of microorganisms to produce biofilm depends on their ability to secrete molecules intrinsic to the organism itself and this... does not determine soil structuring. Please remove.
A: We understand this could be misunderstood. Our idea was to link that the strains producing biofilm may help to the soil aggregation, which structure the soil. However, we have applied some changes to avoid confusions. Thanks!
2.6 In microcosms, is the inserted soil that was sampled directly on site? Or was it generated by a mixture of factors?
A: We have change this in the text to avoid confusions thanks to a previous comment. In brief, we decide to use the less aggregated kind of soil available (a sandy soil) in order to detect any increase in the aggregation cause by our treatments. We were not authorized to use natural soil in the amounts we need to collect for preservation reasons (by legislation). So, we opted for this approach as the most restrictive and similar to the conditions of burnt soil (and available).
Figure 2. I like figure 2. only the two conditions are practically mirror-image except that in one of them there is ash. in section c in the first panel is w/o ash, does it stand for with or without? in which case the grey panel could be removed and specified that the number of treatments is 8 (n=8).
A: We have received different comments about this. The option proposed is more similar to our first way, but we feel that this scheme is clearer, so we prefer to leave as it is. However, we have applied some comments suggested here in order to avoid confusions. Thanks!
L.375-377 sentence very long… not clear.
A: True, we have improved this phrase and this section. Thanks!
L.379 remove During the test and start with “Every”
A: Done! Thanks!
Figure 5. Please improve the quality of image and also increase the size of the letters (N, S…etc)
A: Done, thanks!
3.4 since you chose these specific strains for soil recovery, did you also evaluate how the population varied over time after inoculation? by CFU?
A: Good question. Indeed, we did it. Despite we even were able to visually identify the different strains, the numbers were very variable so we couldn’t conclude with precision enough their evolution, so we considered this information was not relevant enough and become some chaotic element. We have decided to prepare other alternatives to better understand this topic, as label the strains with different fluorescent proteins. In this study we considered more relevant to validate if a consortium can get some kind of results. Later, our idea is to polish the treatment, as well as to include some molecular approach or change the strains to evaluate a better performance for the next work. However, we consider for this work is enough, but we want to improve this for future approaches and quality assessments. Hope this is enough.
L.851-872 I'm confused, what does this part have to do here? shouldn't it go in the soil aggregation part? so 3.5.2. Please change.
A: Yes, this is an editing mistake. In previous versions was in the right place, not sure how this ended here. So sorry, and thanks!
L.874 “weather condition” Climate change.
A: Done! However, following a later comment, we have decided to shortcut this section. Thanks!
L.877-878 “whose pop…” is redundant
A: Same as before, thanks for the indication!
L.887-889 The discussion is well written but it is too long, we must focus more on the discussion of the significant results rather on why this study was made, which, instead, should be written better in the final part of the introduction.
A: True, we have applied this suggestion. Thanks!
L955 “another aspect…” Remove because you don’t tell nothing about other aspect. In our study.
A: Done, thanks!
L.956 Once again, these explanations (thank you because now it is clearer the intent of the research) do not go in this section, but in the introductory part so as to read the work ready for what you expect (things you discover during the reading).
A: Make sense. Hope you can understand that the work is very dense and we have received contradictory comments and indications by other reviewers that preferred this here. I understand as well as you that this is more introduction, but we tried to cope with all revisions as best as we could. Any case, we have applied changes to make part of the introduction the most ‘introductory’ concepts. Thanks at all!
L.981 which micrographs?
A: Figure 7. We were not sure how to call them, some reviewers called them micrographs, but we are not sure about the term as well. We are going to change by ‘pictures’ to avoid confusions. Thanks!
L.990-1005 This part, instead, seems a conclusion and future prospects.
A: Thanks for the comment! Yes, it’s partly, but we consider necessary to discuss the most novel approaches in terms of using microbiota in biotech solutions. It’s indeed very long and most like a conclusion, so we have rearranged it.
There is a redundancy of "in this study". Try to define what has been discovered more directly (maybe you can add here the future perspective – see above).
A: Done! We followed this comment and merge with the main of some other to improve this section. Thanks!
L.1022 “soil fixation” doesn’t mean nothing!!
A: Thanks for the comment! There are not consensual terms for this kind of phenomena, so we decided to use the most common term in Spanish. The translation may not mean the same or cause confusion, so we have decided, following your advice, to change it by ‘soil stabilization’ or ‘soil recovery’ depending on each particular case.
Enlish
A: Hope you understand we are not native and this aspect is always a little difficult to correctly address. However, we have kidnly ask for two native colleagues to help us with this, and we have applied extensive corrections in order to enhance this aspect. Hope this is ok. Thanks!
Reviewer 3 Report
Dear Authors,
Thank you for the modifications you made,
The article is more suitable for publication in its present form,
Unfortunately, I have just one concern, when you answered my comments some of your answers doesn't appear in the new version of the manuscript.
Example all references that in your answers you mentioned you did add them (example: Ref: Heavy Metals Toxicity and the Environment ) do not appear in the text.
Please verify that all your answers are mentioned in the manuscript.
Best Regards,
Author Response
We really understand some of the points were not explicitly included. However, the excess of cites in this paper is not allowing us to include those that are a little out of the main topic. On the other hand, some of them are still on debate with the editors, so may be included. We really appreciate your comments and suggestions, they made this work better and more solid. Thanks!
Round 3
Reviewer 2 Report
Dear authors,
I am pleased with the revisions made and the responses to the various comments I have left you.
Good work.
Author Response
This revision was very enlightening for us, we really appreciate your time and dedication. It's very important for us, and we really think it has improved our work a lot. My best wishes!